# Composition of Clean Marine Air and Biogenic Influences on VOCs during the MUMBA Campaign

**Élise-Andrée Guérette** [1,2,*], **Clare Paton-Walsh** [1], **Ian Galbally** [1,2], **Suzie Molloy** [2], **Sarah Lawson** [2], **Dagmar Kubistin** [1], **Rebecca Buchholz** [1,3], **David W.T. Griffith** [1], **Ray L. Langenfelds** [2], **Paul B. Krummel** [2], **Zoe Loh** [2], **Scott Chambers** [1,4], **Alan Griffiths** [1,4], **Melita Keywood** [2], **Paul Selleck** [2], **Doreena Dominick** [1], **Ruhi Humphries** [1,2] and **Stephen R. Wilson** [1]

1 Centre for Atmospheric Chemistry, University of Wollongong, Northfields Avenue, Wollongong, NSW 2522, Australia
2 Climate Science Centre, CSIRO Oceans and Atmosphere, Aspendale, VIC 3195, Australia
3 Atmospheric Chemistry Observations & Modeling Laboratory, National Center for Atmospheric Chemistry, Boulder, CO 80305, USA
4 ANSTO Institute for Environmental Research, Lucas Heights, NSW 2234, Australia
* Correspondence: elise-andree.guerette@csiro.au; Tel.: +61-3-9239-4484

**Abstract:** Volatile organic compounds (VOCs) are important precursors to the formation of ozone and fine particulate matter, the two pollutants of most concern in Sydney, Australia. Despite this importance, there are very few published measurements of ambient VOC concentrations in Australia. In this paper, we present mole fractions of several important VOCs measured during the campaign known as MUMBA (Measurements of Urban, Marine and Biogenic Air) in the Australian city of Wollongong (34°S). We particularly focus on measurements made during periods when clean marine air impacted the measurement site and on VOCs of biogenic origin. Typical unpolluted marine air mole fractions during austral summer 2012-2013 at latitude 34°S were established for $CO_2$ (391.0 ± 0.6 ppm), $CH_4$ (1760.1 ± 0.4 ppb), $N_2O$ (325.04 ± 0.08 ppb), CO (52.4 ± 1.7 ppb), $O_3$ (20.5 ± 1.1 ppb), acetaldehyde (190 ± 40 ppt), acetone (260 ± 30 ppt), dimethyl sulphide (50 ± 10 ppt), benzene (20 ± 10 ppt), toluene (30 ± 20 ppt), $C_8H_{10}$ aromatics (23 ± 6 ppt) and $C_9H_{12}$ aromatics (36 ± 7 ppt). The MUMBA site was frequently influenced by VOCs of biogenic origin from a nearby strip of forested parkland to the east due to the dominant north-easterly afternoon sea breeze. VOCs from the more distant densely forested escarpment to the west also impacted the site, especially during two days of extreme heat and strong westerly winds. The relative amounts of different biogenic VOCs observed for these two biomes differed, with much larger increases of isoprene than of monoterpenes or methanol during the hot westerly winds from the escarpment than with cooler winds from the east. However, whether this was due to different vegetation types or was solely the result of the extreme temperatures is not entirely clear. We conclude that the clean marine air and biogenic signatures measured during the MUMBA campaign provide useful information about the typical abundance of several key VOCs and can be used to constrain chemical transport model simulations of the atmosphere in this poorly sampled region of the world.

**Keywords:** volatile organic compounds; VOCs; air quality; clean marine air; biogenic; MUMBA

## 1. Introduction

The strong influence of biogenic volatile organic compounds (VOCs) on urban air quality was first pointed out following a study of photochemical smog in Atlanta [1]. As evidence of the adverse

health effects of poor air quality grew, e.g., [2], tougher legislation was passed in North America and Europe, which led to significant reductions in anthropogenic VOC emissions and ozone precursors. As anthropogenic emissions have decreased, the relative importance of biogenic VOCs has grown, with increased interest in the complex chemistry at the urban interface [3]. With reduced emissions in Europe, the focus shifted to mega-cities in China [4] and North America [5]. Intensive measurement campaigns have also been undertaken in Mexico City [6–8] and London [9].

Wilson et al. [10] showed that despite decreases in known ozone precursors between 1996 and 2005, rural ozone concentrations in Europe showed a slight increase. Thus, the importance of understanding background concentrations as well as pollution sources was recognised. This is further exacerbated by the limited knowledge about the diversity of species contributing to atmospheric VOCs [11]. One driver of changes in background concentrations is the long-range transport of pollutants [12], with Asian emissions being transported to the USA [13], and North American emissions impacting western Europe [14].

Within the Australian region, there have been a number of air quality studies, including some specifically aimed at testing the Australian Air Quality Forecasting System [15] in Sydney [16] and Melbourne [17]. The primary focus of these studies was testing the prediction of ozone levels in the urban environment [18]. There have also been Australian campaigns focused on understanding aerosol formation and composition in the urban environment, e.g., [19–21]; coastal environments [22–24]; and within eucalypt forests [25,26]. Recent modelling studies have highlighted the importance of biogenic emissions for ozone formation in the Sydney region [27,28]. In their source apportionment study, Nguyen Duc et al. [27] found that biogenic emissions are the main contributor to ozone levels in the greater Sydney region all year round. Similarly, Utembe et al. [28] found that removing biogenic emissions in their model removed all ozone episodes during extreme heat episodes. Biogenic emissions also influence $PM_{2.5}$ levels in the region, as evidenced by the modelling source attribution study of Chang et al. [29]. Also, data from the Sydney Particle Study [21] indicate that up to 70% of the organic matter fraction of $PM_{2.5}$ aerosol in Sydney in summer is of biogenic origin [30].

Despite their documented impact on air quality, biogenic emissions in this region are still poorly understood. Commonly used biogenic emission models such as the Model of Emissions of Gases and Aerosols from Nature (MEGAN) [31] have been found to perform poorly over south-east Australia [32]. Recent modelling work [33,34] has highlighted potential issues relating to the light-dependency (or lack thereof) of monoterpene emissions and to the sensitivity of isoprene emissions to drought conditions.

There are few existing measurements of VOC mole fractions in the background clean marine air of the Southern Hemisphere. Most of the existing data come from ship cruises and aircraft campaigns. In Australia, there are some background VOC measurements available from the Cape Grim Baseline Air Pollution Station [35–37], which is located on the north-western tip of Tasmania ($-40.683°$, $144.689°$) and is part of the World Meteorological Organisation's (WMO) Global Atmosphere Watch (GAW) network of stations (http://www.bom.gov.au/inside/cgbaps/), however such stations are separated by large distances in the Southern Hemisphere and further measurements at different latitudes are required to build a more complete picture of VOCs in the region. In addition, there is a paucity of measurements of terrestrial VOCs in Australia and evidence that biogenic VOCs may be poorly represented by atmospheric chemical transport models in this region [32]. Hence, there is a need for additional observations of VOCs in both marine and terrestrial air masses in Southern Hemisphere mid-latitudes.

In this paper, we present VOC measurements from a campaign known as MUMBA (Measurements of Urban, Marine and Biogenic Air). Previous papers have presented a detailed description of the MUMBA campaign [38]; explored the urban air quality observed during the campaign [39]; the general characteristics of the aerosols observed [40] and new particle formation events that occurred [41]. Further implications of the campaign measurements for air quality have been explored by use of MUMBA as a case study for an extensive air quality modelling intercomparison exercise (e.g., [28,42–45]). Here we describe in detail the measurements of VOCs made during periods when clean marine air was transported to the site, and when the VOCs sampled were predominantly of biogenic origin.

## 2. Methods

### 2.1. The MUMBA Campaign

The MUMBA campaign took place over an 8-week period in the Australian city of Wollongong, New South Wales (NSW), from 21 December 2012 to 15 February 2013. A total of 20 instruments were deployed, with the majority located at the eastern campus of the University of Wollongong (34.397° S, 150.900° E). Ambient air was sampled approximately 10 m above ground level from a dedicated mast fitted with several air intakes. The main instrumentation relevant for this study was a proton transfer reaction mass spectrometer (PTR-MS) for measurement of volatile organic compounds (VOCs) and a Fourier transform infrared spectrometer (FTIR), for measurement of CO, $CO_2$, $CH_4$ and $N_2O$.

The campaign site was impacted by different influences depending on wind direction, with a steep forested escarpment to the west; ocean to the east; and a major industrial area (Port Kembla) to the south [38,39]. See Figure 1 for maps of the region and of the direct vicinity of the MUMBA site.

Relevant to this study is the clean marine air that can usually be sampled during south-easterly winds and the strong biogenic influences that are experienced during westerly winds. The MUMBA site is located in a grassy field with a suburban main road ~125 m to the east, beyond which is a strip of forested parkland ~300 m wide (Puckey's Estate) before the beach and the ocean. Dense eucalypt forest lies beyond the escarpment ~3 km directly to the west with forested regions significantly closer to the site in the north-west direction (~2.5 km) than to the south-west (~10 km). See Figure 1b for a local map. A more detailed description of MUMBA and the urban air quality during the campaign are available in the literature [38,39], and the data are published in PANGAEA [46].

Also of relevance to this study, are the radon-222 measurements made by ANSTO (Australian Nuclear Science and Technology Organisation) at Warrawong, NSW, roughly 10 km south of the MUMBA site and 3 km inland from the coast. Continuous, direct, hourly atmospheric radon concentration measurements were made using a 1500 L dual flow loop, two-filter radon detector [47,48] sampling at a height of 2 m above ground level. The location of this site is indicated on the map in Figure 1a.

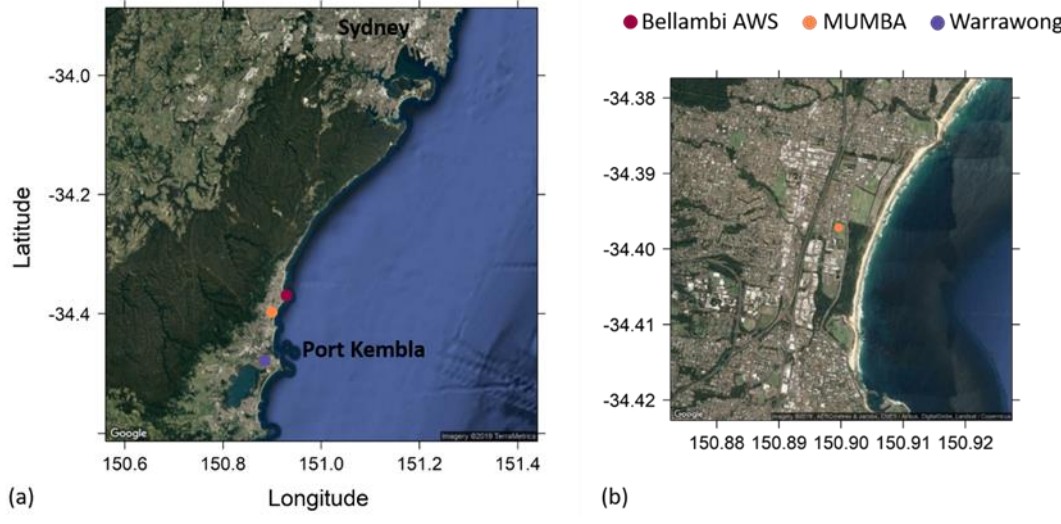

**Figure 1.** Maps showing (**a**) regional influences on the MUMBA site, including Sydney and Port Kembla as well as the location of the Warrawong radon measurement site operated by ANSTO and of the Bellambi Automated Weather Station (AWS) operated by the Australian Bureau of Meteorology (BOM) and (**b**) local influences on the MUMBA site, including Puckey's Estate.

### 2.2. The Proton Transfer Reaction-Mass Spectrometer (PTR-MS)

#### 2.2.1. PTR-MS Operation

A commercial PTR-MS (IONICON) equipped with a quadrupole mass spectrometer was used in this study along with custom-built auxiliary equipment [35,49] that allows automated calibration and zero measurements. Zero measurements were performed twice daily for 40 min each time (at 00:50 and at 15:00) by sampling ambient air that had been stripped of VOCs by passing through a platinum-coated glass wool catalyst heated to 350 °C. A multi-species, single-point calibration was performed daily (from 01:30 until 03:00) by introducing a known flow of calibration standard into the zero-air stream. Calibration mole fractions were ~10 to 20 ppb for each species present in the standard. Calibration is based upon two cylinders: CC345925, of standard gas mixture of 15 VOCs at nominal levels of ~1 ppm from Apel-Riemer Environmental Inc. (Broomfield, CO, USA) and ALM25971, of standard gas mixture of 2 VOCs at nominal levels of ~1 ppm from Air Liquide–Scott Specialty Gases (Plumsteadville, PA, USA).

The PTR-MS was programmed to scan through its range of mass-to-charge ratios (m/z) with a dwell time of one second, for a total cycle time of about 3 min. For a few days of the campaign, however, the dwell time for m/z 56–190 was set to 100 milliseconds accidentally, leading to a more rapid cycle time of approximately 50 s for this period. The instrument ran continuously over the campaign; gaps in the data are due to ion voltage issues or to excessive heat in the hut that housed the instruments (that forced a protective shut down of all instruments).

The instrument sampled outside air from a dedicated 15 m, $\frac{1}{4}$ in. Teflon line at a flow rate of 1.4 L/min. The inlet was capped with an inverted Teflon cup. This system has been tested for VOC losses by introducing various VOC mixtures in the ppb range to the PTR-MS both via the inlet line and directly and then comparing the measured signals. Using this method, the upper bound to the inlet losses was estimated to be <5%. As a single inlet was used during the entire campaign, it is possible that VOCs adsorbed to the inlet wall during sampling of polluted conditions may have desorbed during sampling of clean conditions and contributed a small error to the measurements.

#### 2.2.2. PTR-MS Data Processing and Detection Limits

The time series of raw counts obtained during the campaign were separated into ambient, zero and calibration data using measurement timestamps. As the instrument signal is proportional to the reagent ion concentration, the signal was normalised to $10^6$ reagent ion counts to remove the effect of any ion source fluctuations over the campaign period. Ambient data were zero corrected using interpolated zero values (for every measurement, the interpolated zero value is the average of the zero periods immediately preceding and following it), whereas the calibration data were zero corrected using the zero period immediately preceding the calibration period.

The zero-corrected calibration data were then used to derive calibration factors for the m/z ratio of each species present in the calibration standards. The response of the instrument to certain species in the calibration standards varied depending on the amount of water (in g m$^{-3}$) present in ambient air at the time of the calibration. For these species, water-dependent calibration factors were derived. The calibration factors were then applied to the zero-corrected ambient data to obtain ambient mole fractions. For some m/z, the ambient signal was corrected for known interferences before a calibration factor was applied: m/z 31 (formaldehyde) was corrected for interference from methyl hydroperoxide (m/z 31 - 0.9* m/z 49) [50] and m/z 79 (benzene) was corrected for interference from acetic acid clusters (m/z 79 - 0.05*m/z 61) [51]. The VOC measurements presented here are in the commonly used units of parts per billion (ppb) and parts per trillion (ppt) which for an ideal gas (as probably occurs at the extreme dilutions present here) are equivalent to the SI unit of mole fraction, nanomole/mole and picomole/mole, respectively.

The detection limit for compounds for which a calibration standard was available was inferred using principles from ISO 6879. Briefly, for each zero period, an average zero signal for every m/z was calculated as well as the deviation from the average of each zero scan. The 95th

percentile of all the deviations for each mass is the detection limit. Detection limits for longer periods (e.g., hourly averages) were determined, assuming Poisson statistics, by dividing the single measurement (1 s or 100 milliseconds) detection limit by the square root of the number of measurements in the longer measurement period. For a one second dwell time there are 20 measurements per m/z per hour. This can be used to determine detection limits at other intervals from the hourly data provided. Detection limits in normalised counts were converted to mole fractions using average calibration factors. Detection limits varied over the campaign, mostly due to varying instrument conditions (e.g., ion source voltage and dwell time). The poorest detection limits calculated for a 1-h sampling period are reported in Table 1, along with the calibration factors for each species (sensitivities, number of counts per second per part per billion of each gas ncps/ppb). Note that although m/z 71, 107, 121 and 137 were calibrated using the single species listed in Table 1, for ambient measurements these m/z represent sum measurements of methacrolein and methyl vinyl ketone (MACR + MVK); $C_8H_{10}$ aromatic species; $C_9H_{12}$ aromatic species and monoterpenes, respectively. Other m/z are interpreted as single species, although interference from other unidentified compounds at the same m/z cannot be excluded.

**Table 1.** Summary of PTR-MS sensitivities (ncps/ppb) and detections limits (ppb) for 13 species during the MUMBA campaign. NA means that the calibration was not water dependent. Not determined means that water dependence was not tested. Water is in units of g m$^{-3}$.

| Protonated Mass (m/z) | Species | Sensitivity (ncps/ppb) | | Detection Limit for a 1-h Measurement (ppb) |
| --- | --- | --- | --- | --- |
| | | Water Dependency | Average | |
| 31 | Formaldehyde | $4.6 - 0.11*[H_2O]$ | 2.9 | 0.205 |
| 33 | Methanol | $5.9 + 0.7*[H_2O]$ | 16.7 | 0.062 |
| 42 | Acetonitrile | $21.5 + 1.4*[H_2O]$ | 43.6 | 0.002 |
| 45 | Acetaldehyde | $21.4 + 1.3*[H_2O]$ | 41.7 | 0.018 |
| 59 | Acetone | $25.1 + 1.9*[H_2O]$ | 54.5 | 0.013 |
| 63 | Dimethyl sulphide | Not determined | 36 | 0.010 |
| 69 | Isoprene | Not determined | 31.4 | 0.005 |
| 71 | Methacrolein | $25.9 + 1.9*[H_2O]$ | 55.2 | 0.005 |
| 79 | Benzene | NA | 23.9 | 0.012 |
| 93 | Toluene | NA | 31.1 | 0.008 |
| 107 | m-xylene | NA | 34.9 | 0.016 |
| 121 | 1,3,5-trimethyl benzene | NA | 32.8 | 0.013 |
| 137 | $\alpha$-pinene | NA | 15.7 | 0.016 |

## 2.3. Fourier Transform Infrared Spectrometer (FTIR)

A FTIR trace gas analyser measured carbon dioxide ($CO_2$), methane ($CH_4$), carbon monoxide (CO) and nitrous oxide ($N_2O$) throughout the campaign. The analyser consists of a 1 cm$^{-1}$ resolution Bruker IR Cube Fourier transform infrared spectrometer with a calcium fluoride beamsplitter, which modulates the infrared beam before it enters a multi-pass cell to produce a total path length of 24 m. The beam is then focused onto a thermoelectrically-cooled mercury cadmium telluride detector. Spectra are analysed using the HITRAN database [52] and the non-linear least-squares program MALT (Multiple Atmospheric Layer Transmission [53,54] in the following spectral regions:

1. 2097–2242 cm$^{-1}$, optimised for $N_2O$ and CO, also fitting $CO_2$;
2. 2150–2310 cm$^{-1}$, fitting CO, $N_2O$, $H_2O$ and $CO_2$ isotopologues;
3. 3001–3150 cm$^{-1}$, fitting $CH_4$ and $H_2O$;
4. 3520–3775 cm$^{-1}$, fitting $CO_2$ and $H_2O$.

The resulting measurements of mole fractions of $CO_2$, CO, $N_2O$ and $CH_4$ meet the precision and accuracy requirements set by WMO GAW standards for baseline air [54,55].

### 2.4. Identifying Episodes of Clean Marine Air at the MUMBA Site

The lowest mole fractions of most trace gases were measured when winds were from the south-east. These winds bring air from the ocean and can potentially be used to characterise clean marine air at this latitude (34°S). Wollongong is more than 800 km from the nearest WMO GAW station (Cape Grim, Tasmania), and so it is useful to characterise the atmospheric composition of the clean marine air arriving at the MUMBA site.

To identify periods of marine air, we used the radon measurements made by ANSTO in Warrawong. Radon is an unambiguous indicator of recent terrestrial influences on an air mass, due to a combination of its half-life of 3.82 days and the fact that radon is emitted with a terrestrial flux over 500 times greater than the oceanic flux: Australian radon emissions average 23 mBq m$^{-2}$ s$^{-1}$ from land [56], whereas mean oceanic emissions are estimated to be 0.04 mBq m$^{-2}$ s$^{-1}$ [57]. Here we select radon levels below 200 mBq m$^{-3}$ to reflect oceanic air. This threshold is double that commonly utilised at Cape Grim for identifying clean marine air [58,59] and four times higher than that of pristine oceanic air [60], but much less than the several thousand mBq m$^{-3}$ observed in continental air. Note that the sampled marine air will have accumulated minor amounts of radon during the overland transport from the coast to the ANSTO monitoring site (approximately 3 km inland). Radon levels below 200 mBq m$^{-3}$ were recorded for 11 consecutive hours on 26 December 2012, 6 h on 5 February 2013, 6 h on 6 February 2013 and 4 h on 13 February 2013. Back trajectory analysis also indicates that the air masses arriving at Wollongong during these periods had travelled over the ocean for the previous 96 h (see Appendix A, Figure A1). Local meteorological conditions during these periods are summarised in Table 2. Meteorological conditions were remarkably stable during all marine air periods, including the relatively long 11-h period on 26 December 2012, with temperature fluctuating by less than a degree Celsius. Both 26 December 2012 and 13 February 2013 were mostly cloudy days, whereas it was sunny on 5 February and 6 February 2013.

**Table 2.** Summary of local meteorological conditions during the four periods that marine air—as characterised by low radon levels (<200 mBq/m$^3$) for four or more continuous hours—reached Wollongong during the MUMBA campaign.

| Parameters | Period | | | |
|---|---|---|---|---|
| | December 26th 08:00–18:59 | February 5th 12:00–17:59 | February 6th 13:00–18:59 | February 13th 14:00–17:59 |
| Wind direction (°) | 159 ± 4 | 70 ± 20 | 49 ± 9 | 160 ± 10 |
| Wind speed (m s$^{-1}$) | 5.0 ± 0.7 | 3.0 ± 0.6 | 5.6 ± 0.5 | 3.3 ± 0.6 |
| Temperature (°C) | 20.0 ± 0.5 | 23.4 ± 0.3 | 24.2 ± 0.6 | 23.1 ± 0.3 |
| Relative humidity (%) | 65 ± 3 | 76 ± 2 | 70 ± 3 | 68 ± 1 |
| Radon (mBq/m$^3$) | 150 ± 20 | 120 ± 20 | 150 ± 30 | 150 ± 30 |

Local winds on 26 December and 13 February were from the south-east, whereas they were from the north-east on 6 February. On 5 February, local winds shifted direction from east-south-east to north-east during the marine air period and the mole fractions of many trace gases increased steadily as the wind shifted direction. North-easterly winds, the dominant summer sea breeze direction in Wollongong, can transport pollution from the Sydney basin to the Wollongong region [61]. For this reason, only the two low radon events with south-easterly winds (the main episode on 26 December 2012 and the shorter episode on 13 February 2013) are considered to be episodes of clean marine air arriving at the MUMBA site.

Mean mole fractions were calculated from hourly averages for thirteen VOC species, CO, NO$_x$ and ozone; these are presented in Table A1 (Appendix A), along with the standard deviation of the mean. The standard deviation is given as an indication of variability in the mole fractions over the periods and is not meant to indicate error in the measurements.

## 3. Results

### 3.1. Ambient VOCs Measured during the MUMBA Campaign

This paper focuses on periods when clean marine air was transported to the site, and when the VOCs sampled were predominantly of biogenic origin; however, as there is a relative paucity of VOC measurements in Australian cities, we provide statistics that describe the mole fractions of VOCs that were observed during the entire MUMBA campaign in this section. Table 3 lists the mean and standard deviation of mole fractions for all VOCs quantified during the MUMBA campaign. The range and statistics for the quartile ranges are also provided.

**Table 3.** Mole fractions of VOCs and other gases (in ppb) measured during the MUMBA campaign; total number of hours of measurement; mean; standard deviation of the mean; and 1st, 2nd (median) and 3rd quartiles and total range of measured mole fractions.

| Mass | Species | Number of Hours | Mean (ppb) | SD (ppb) | 1st Quartile (ppb) | Median (ppb) | 3rd Quartile (ppb) | Range Min–Max (ppb) |
|---|---|---|---|---|---|---|---|---|
| 31 | Formaldehyde | 1025 | 1.19 | 0.95 | 0.65 | 0.92 | 1.44 | 0.09–8.69 |
| 33 | Methanol | 1025 | 2.2 | 1.6 | 1.3 | 1.8 | 2.6 | 0.7–12.6 |
| 42 | Acetonitrile | 1018 | 0.076 | 0.023 | 0.061 | 0.071 | 0.085 | 0.037–0.217 |
| 45 | Acetaldehyde | 1025 | 0.41 | 0.31 | 0.19 | 0.32 | 0.54 | 0.06–2.44 |
| 59 | Acetone | 1025 | 0.71 | 0.46 | 0.41 | 0.59 | 0.84 | 0.19–3.95 |
| 69 | Isoprene | 1027 | 0.29 | 0.42 | 0.07 | 0.18 | 0.37 | 0.003–4.57 |
| 71 | MACR + MVK | 1025 | 0.20 | 0.35 | 0.05 | 0.09 | 0.20 | 0.006–4.31 |
| 79 | Benzene | 1027 | 0.113 | 0.094 | 0.049 | 0.087 | 0.149 | 0.004–0.816 |
| 93 | Toluene | 1027 | 0.31 | 0.34 | 0.09 | 0.20 | 0.37 | 0.004–2.67 |
| 107 | $C_8H_{10}$ | 1015 | 0.24 | 0.27 | 0.07 | 0.14 | 0.30 | 0.005–2.11 |
| 121 | $C_9H_{12}$ | 1027 | 0.15 | 0.15 | 0.06 | 0.10 | 0.17 | 0.007–1.71 |
| 137 | Monoterpenes | 1027 | 0.12 | 0.16 | 0.04 | 0.07 | 0.14 | 0.004–1.39 |
| n/a | Carbon monoxide | 1139 | 113 | 86 | 65 | 85 | 122 | 46–860 |
| n/a | Ozone | 1284 | 18.3 | 8.7 | 12.9 | 18.2 | 23.2 | 1.0–53.9 |
| n/a | NO | 1237 | 2.4 | 6.8 | 0.3 | 0.9 | 2.1 | 0–136.4 |
| n/a | $NO_2$ | 1237 | 5.0 | 4.0 | 1.8 | 3.8 | 7.2 | 0.2–23.1 |
| n/a | NOx | 1237 | 7.5 | 9.3 | 2.6 | 4.9 | 9.1 | 0.2–156.2 |

### 3.2. Comparison of Clean Marine Air at MUMBA with Measurements at Cape Grim, Tasmania

Table 4 shows a comparison of mean mole fractions of five trace gases measured during the episode on 26 December 2012 with data from Cape Grim, Tasmania. Cape Grim baseline values for $CO_2$, CO, $CH_4$ and $N_2O$ were extrapolated to 26 December 2012 using a curve-fitting algorithm [62] applied to the long-term baseline flask record. This tracks the seasonal cycle with an 80-day smoothing and is calculated for daily intervals. Monthly means are calculated as the mean of daily values from the smoothed curves. The differences between 26 December and December mean values for $CH_4$ and CO are due to significant changes over the course of the month related to their seasonal cycles.

It is clear from the comparison that the MUMBA site may experience episodes of clean marine air with similar characteristics to that sampled at Cape Grim under baseline conditions. Mole fractions of the greenhouse gases $CO_2$, $CH_4$ and $N_2O$ measured at the MUMBA site during the 26 December clean marine air episode (391.0 ± 0.6 ppm, 1760.1 ± 0.4 ppb and 325.04 ± 0.08 ppb respectively) are very similar to the baseline values extrapolated for the same day at Cape Grim (391.09 ppm, 1760.1 ppb and 324.96 ppb); however, the relatively large standard deviations in the MUMBA measurements indicate atmospheric variability. Figure A2 (Appendix A) shows the 3-min timeseries for $CO_2$, $CH_4$ and $N_2O$ and CO between 08:00 and 18:59 on 26 December. The dashed horizontal line indicates the Cape Grim baseline value. The figure shows that $CO_2$ was slightly below baseline for much of the day, probably due to photosynthesis from the local vegetation (e.g., Puckey's Estate). $CO_2$ mole fractions then start to increase in the late afternoon, as wind speed starts decreasing. $CH_4$ mole fractions were on baseline for much of the day, except for some short-lived episodes above and below. $N_2O$ was on

baseline until just after 14.00, when, presumably, light precipitation caused a release of $N_2O$ from the land surrounding the measurement site.

The average CO mole fraction at the MUMBA site during the 26 December clean air episode was 52.4 ± 1.7 ppb, ~4–6 ppb above the December Cape Grim baseline values of 45.9–48.6 ppb. Note that this difference may be larger by an additional 2 ± 1 ppb due to small drift in the FTIR CO calibration tanks over time. Again, the large standard deviation in the MUMBA CO measurement indicates atmospheric variability. The bottom panel of Figure A2 in Appendix A reveals many short-lived spikes in the 3-min FTIR record on the day. This might be attributed to local traffic on the road east of the site. The higher background may also be due to local influences (there is a major shipping corridor just off the coast of Wollongong), but there is also an expected latitudinal gradient in CO due to mixing of more polluted Northern Hemisphere air across the chemical equator [63,64]. The CO mole fractions observed on 26 December 2012 at the MUMBA site are consistent with clean air values determined from a multi-year record (2011–2014) at the nearby University of Wollongong [65].

Mean $O_3$ levels during this time were 20.5 ± 1.1 ppb, slightly above the campaign average of 18.3 ppb. This is likely due to minimal loss of $O_3$ *via* titration by NO, as a consequence of low NO values (0.3 ± 0.1 ppb, compared to a campaign average value of 2.4 ppb). $NO_2$ and total $NO_X$ were also low for Wollongong during the 26 December clean air episode, with $NO_2$ values averaging 0.7 ± 0.2 ppb and $NO_X$ values averaging 1.0 ± 0.3 ppb, compared to campaign averages of 5 ppb and 7.5 ppb respectively.

**Table 4.** Comparison of hourly averaged trace gas mole fractions (mean ± standard deviation) in clean marine air sampled during the MUMBA campaign with Cape Grim Baseline record for the equivalent time period.

| | MUMBA 26 December 2012 (08:00 to 18:59) | Cape Grim Baseline 26 December 2012 | Cape Grim Baseline December 2012 |
|---|---|---|---|
| $CO_2$ (ppm) | 391.0 ± 0.6 | 391.09 | 391.16 ± 0.07 |
| $CH_4$ (ppb) | 1760.1 ± 0.4 | 1760.1 | 1763.7 ± 3.4 |
| $N_2O$ (ppb) | 325.04 ± 0.08 | 324.96 | 325.1 ± 0.2 |
| CO (ppb) | 52.4 ± 1.7 | 45.9 | 48.6 ± 2.5 |
| $O_3$ (ppb) | 20.5 ± 1.1 | | 20 ± 4 * |

* not baseline filtered.

### 3.3. Comparison of VOCs in Clean Marine Air at MUMBA with other Measurements in the Literature

The average VOC mole fractions observed for the periods when clean marine air reached Wollongong on 26 December 2012 and 13 February 2013 are shown in Table 5, and compared with values from the literature. Biogenic species and their oxidation products are excluded because of the contaminating influence of Puckey's Estate to the east of the site [38]. In Table 5, two of the literature sources relate to VOC measurements made at the Cape Grim Baseline Air Pollution Station [35,36], two others present aircraft measurements [66,67] and nine describe ship-based measurements of mole fractions above the ocean surface in various locations [68–76]. When comparing to clean marine air (Table 5), it should be noted that Table 5 lists values in parts per trillion, whereas Table 3 lists values in parts per billion.

**Table 5.** Average mole fractions and standard deviation (in brackets) of selected compounds in marine air during MUMBA, and in marine air in other parts of the Southern Hemisphere and the tropics. For the other studies, a range of values is presented when possible, with the lowest value being the lowest reported across all the studies included and the highest being the highest reported. All mole fractions are in parts per trillion (ppt) unless otherwise indicated.

| Protonated Mass | Species | MUMBA 26 Dec 2012 | MUMBA 13 Feb 2013 | Cape Grim [35,36] (Southern Ocean) | Southern Hemisphere Mid-Latitude Oceans [68,70,74] | Tropical Oceans [67,71,73,75,76] |
|---|---|---|---|---|---|---|
| 31 | Formaldehyde | 590(80) | 600(130) | – | – | 211(144) |
| 33 | Methanol | 1340(170) | 1020(70) | 476(26)–633 | 546(139)–680(260) | 575(211)–890(400) |
| 42 | Acetonitrile | 56(5) | 65(4) | 25(1)–32 | 23(7)–50(10) | 110(20)–142(20) |
| 45 | Acetaldehyde | 190(40) | 120(30) | <4–53 | 290(100) | 178(30)–204(40) |
| 59 | Acetone | 260(30) | 270(10) | 61–118(5) | 127(38)–453(114) | 361(51)–530(200) |
| 63 | Dimethyl sulphide | 50(10) | 35(8) | ~80–95 | 77(34) | 50(50)–89(58) |
| 79 | Benzene | 20(10) | 28(5) | 4(1) | 10–80(40) | – |
| 93 | Toluene | 30(20) | 45(3) | – | 9 | 9(7) |
| 107 | $C_8H_{10}$ | 23(6) | 70(40) | – | 10 | – |
| 121 | $C_9H_{12}$ | 36(7) | 50(20) | – | – | – |
| | CO (ppb) | 52(2) | 58(6) | ~50 | 45(4)–54(4) | 64(6)–111(16) |

## 3.4. Biogenic VOCs of Terrestrial Origin Measured during MUMBA

The biogenic VOCs measured during this campaign are of interest since there is a paucity of measurements of terrestrial biogenic VOCs in Australian environments [32,33]. The MUMBA site received biogenic influences when the winds were from the east, because these winds bring air over the nearby Puckey's Estate (~150 m distant). The site also received biogenic signatures from the more distant forested escarpment to the west (between 2.5 km and 10 km away depending on wind direction). The times series of isoprene, the sum of methacrolein and methyl vinyl ketone, the sum of monoterpenes, and formaldehyde measured during MUMBA are shown in Figure 2 and those of methanol, acetaldehyde and acetone are shown in Figure 3.

The campaign encompassed two of the hottest days on record for the region [28]. On 8 January 2013, strong north-westerly winds (≥ 5 m s$^{-1}$ through most of the time between 09:00 and 15:00) brought high air temperatures averaging above 40 °C between 14:00 and 17:00. Trace gas measurements showed strong biogenic signals, with isoprene mole fractions nearly an order of magnitude higher than typically measured during MUMBA—peaking with hourly average values in excess of 4 ppb (see Figure 2).

Similar conditions were encountered on 18 January 2013 when west-north-westerly winds > 4 m s$^{-1}$ brought air temperatures peaking with hourly mean values above 44 °C between 12:00 and 13:00. Again, very high levels of isoprene were measured with peak hourly averages over 4 ppb (Figure 2). Peaks were also seen in mole fractions of isoprene oxidation products (the sum of methacrolein and methyl vinyl ketone, MACR +MVK), formaldehyde, methanol, acetaldehyde, acetone and, to a much lesser extent, in monoterpenes (see Figures 2 and 3). Another feature of the time series, associated with peak mole fractions of many of the longer-lived biogenic VOCs (e.g., acetone), is seen on 12 January 2013. Although not an exceptionally hot day in Wollongong, 12 January 2013 was, at the time, the hottest day on record in New South Wales [77].

Westerly winds were very infrequent during the MUMBA campaign and so the influences from Puckey's Estate dominate most of the time, except during the two local extreme heat days when strong westerlies brought biogenic influences from the more distant escarpment. For this reason, the biogenic VOCs on these extreme heat days are discussed separately from the rest of the campaign.

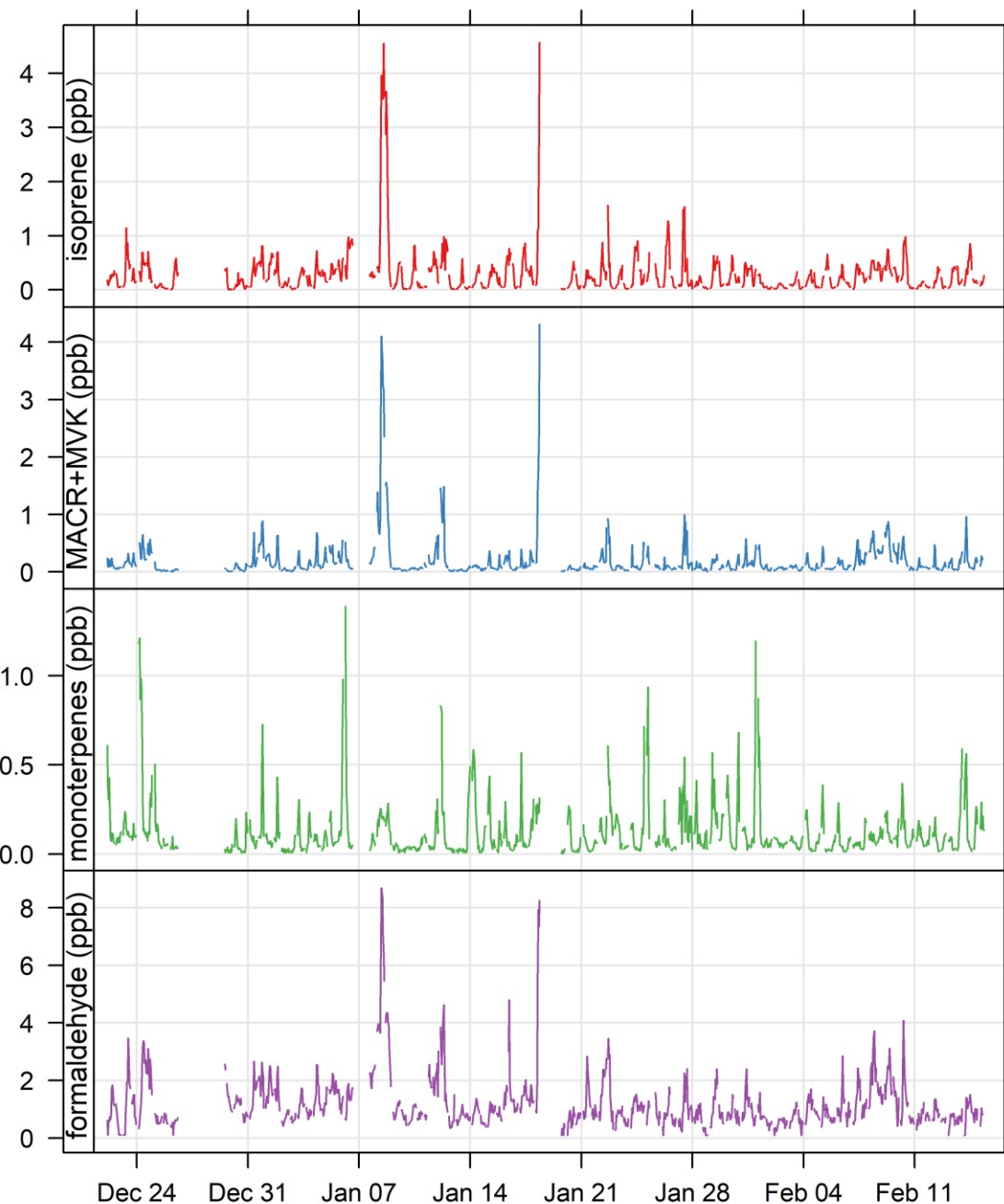

**Figure 2.** Temporal evolution of biogenic VOCs (isoprene, monoterpenes) and their oxidation products (the sum of methacrolein and methyl vinyl ketone, formaldehyde) during MUMBA.

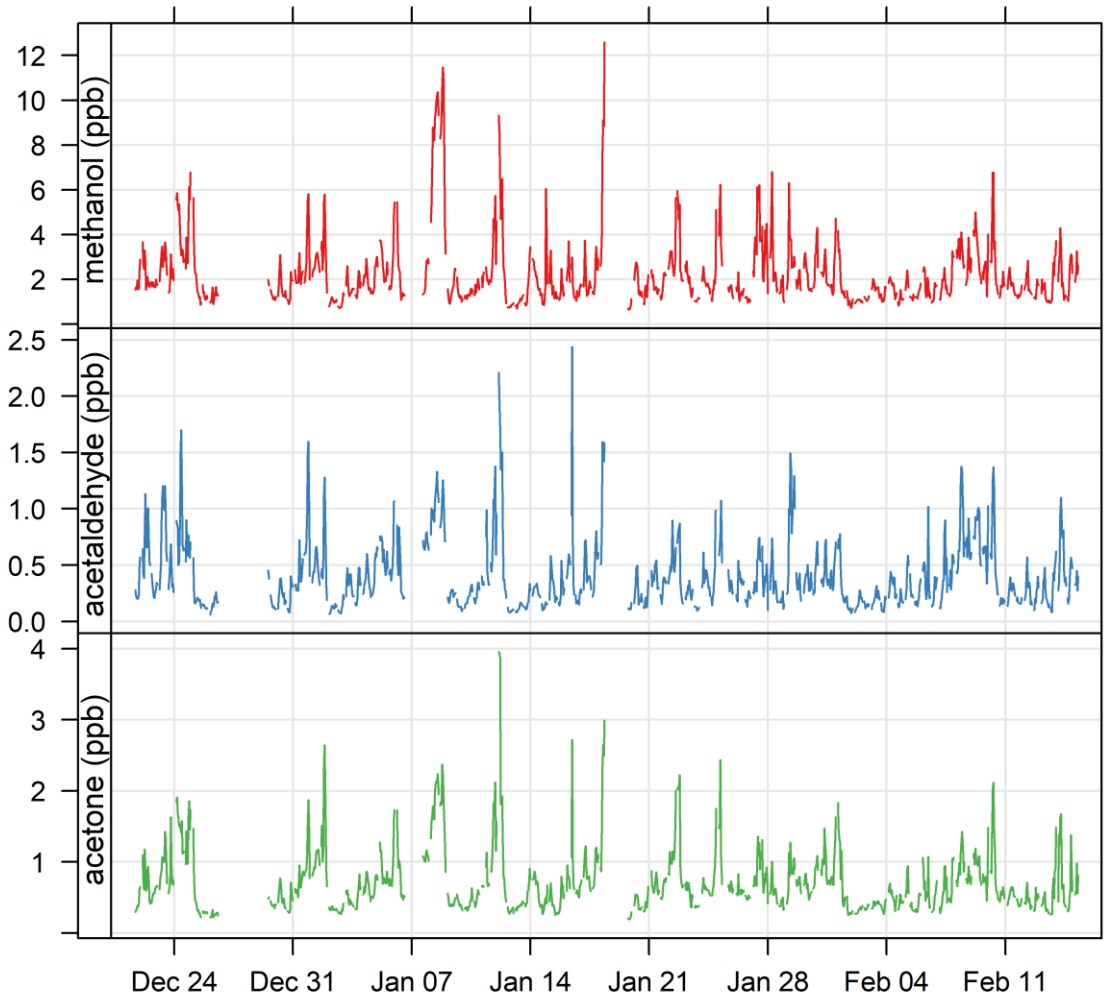

**Figure 3.** Temporal evolution of oxygenated VOC species (methanol, acetaldehyde and acetone) during MUMBA.

## 4. Discussion

### *4.1. VOCs in Clean Marine Air*

#### 4.1.1. Oxygenated VOCs

Methanol is one of the most abundant oxygenated VOCs [78]. It is more or less evenly distributed with altitude in the troposphere over the Pacific Ocean [66,67]. Background values measured at the surface in marine air show a relatively small spread, with 476–633 ppt observed at Cape Grim in summer [35,36] and similar values observed over the Atlantic Ocean both at southern [74] and tropical [75] latitudes, and over both the southern [68] and tropical Indian Ocean [67,76]. Modelled background values [78,79] agree quite well with these observed values in remote marine air. Methanol mole fractions observed in marine air during MUMBA are higher (>1000 ppt) than the ones discussed above (see Table 5), and an influence from Puckey's Estate is suspected, since vegetation is known to be the major source of atmospheric methanol [78,79].

Acetone is also evenly distributed with altitude in the troposphere over the Pacific Ocean [66,67], although slightly lower mole fractions were noted in the Southern Hemisphere [66]. Background values measured at the surface show a latitudinal gradient, as seen in Table 5. This gradient was also observed during a transect of the southern Indian Ocean as part of the MANCHOT campaign [68]. Colomb et al. [68] reported 453 ± 114 ppt in pristine marine air, which is roughly double what was observed in marine air during MUMBA, and nearly four times that observed at Cape Grim in

summer [35,36]. Colomb et al. [68] report diel variations for both acetone and acetaldehyde that correlate well with those of isoprene and the monoterpenes, which they say are consistent with emission from phytoplankton during daylight hours, either directly or through photochemistry. This is in contrast to Williams et al. [74], who report acetone mole fractions of 127 ± 38 ppt under remote marine conditions in the southern Atlantic Ocean, and a similar value (100 ± 16 ppt) under bloom conditions. All the mole fractions of acetone discussed so far have been measured using PTR-MS. Singh et al. [67] used a gas chromatography system aboard an aircraft and reported 466 ± 97 ppt of acetone in the 0–2 km layer of the troposphere in unpolluted air over the tropical Pacific, which is in line with PTR-MS values reported by other groups for surface marine air over tropical oceans [73,75,76]. Marandino et al. [71] reported a mole fraction of 361 ± 51 ppt over the western tropical Pacific using atmospheric pressure chemical ionisation mass spectrometry (API-CIMS).

Acetaldehyde is another oxygenated species that is more or less evenly distributed with altitude in the troposphere over the Pacific Ocean [66,67]. As for acetone, slightly lower mole fractions of acetaldehyde were observed in the Southern Hemisphere than in the Northern Hemisphere [66]. There appears to be a latitudinal gradient in the surface measurements, with the lowest values observed at Cape Grim [35,36] and higher values measured over the tropical oceans. The value reported by Colomb et al. [68] and listed for mid-latitudes in Table 5 (290 ± 100 ppt) is higher than measured during MUMBA (120–190 ppt) and is indeed higher than what they themselves observed at other times during their cruise. They measured mole fractions as low as 120 ± 40 ppt during other parts of their voyage, which support the hypothesis of local biogenic emissions and or photochemical production (discussed for acetone above) on the pristine marine leg of their journey. Modelled background values are in general slightly higher than observed values [80].

Formaldehyde is an oxidation product of isoprene; however, it also has a global background presence due to the atmospheric oxidation of methane. Its mole fraction decreases with altitude in the troposphere over the Pacific Ocean [66,67]. No significant difference between mole fractions in the Southern and the Northern Hemispheres were noted [66]. Background values measured at the surface are quite sparse in the mid-latitudes of the Southern Hemisphere. Our PTR-MS mole fractions in clean marine air (590–600 ppt) are similar to that observed in the Gulf of Mexico (540 ppt) [69], but nearly twice that reported for the 0–2 km layer of the troposphere in unpolluted air over the tropical Pacific [67] (see Table 5). Satellite observations [81] show increased formaldehyde total column amounts near the east coast of Australia; this was also observed in DOAS measurements over the Pacific Ocean (between Japan and Townsville, QLD, Australia) [72]. Peters, et al. [72] also reported a strong (x2) diel cycle in the formaldehyde total column. The influence of Puckey's Estate on our measured mole fractions of formaldehyde in marine air at the MUMBA site cannot be excluded.

### 4.1.2. Dimethyl Sulphide

PTR-MS measurements of dimethyl sulphide in marine air are fairly common, in fact, all of the values reported in Table 5 were obtained using this measurement technique. One study [68] compared their PTR-MS (77 ± 34 ppt) and GC-MS (91 ± 46 ppt) measurements and found reasonable agreement ($y = 0.96x + 37$ (pptv); $r = 0.71$). Dimethyl sulphide is emitted as a result of biological activity in the oceans and plays important roles in atmospheric chemistry and climate through its involvement in the global sulphur cycle [82]. The values reported for marine air during MUMBA are in agreement with those measured elsewhere (Table 5).

### 4.1.3. Acetonitrile

PTR-MS measurements of acetonitrile are also widespread, with all of the values reported in Table 5 being derived using this measurement technique. Although possible interferences at m/z 42 have been identified [83], these should not be significant in marine air. Mole fractions of acetonitrile observed in marine air during MUMBA are higher (56 ± 5 ppt on 26 December 2012 and 65 ± 4 ppt on 13 February 2013) than those measured at Cape Grim (<32 ppt, see Table 5) [35,36].

Colomb et al. [68] observed a latitudinal gradient for acetonitrile between 24°S and 49°S over the Indian Ocean. They reported a mole fraction of 23 ± 7 ppt for what they determined to be pristine marine air (between 45°S and 49°S) and 50 ± 10 ppt between 30°S and 40°S, which is the latitudinal range in which Wollongong is located. Values measured over tropical oceans are even higher than those at mid-latitudes, and this gradient is most likely due to the diminishing influence of biomass burning sources at more southern latitudes, combined with a hypothesised ocean sink [84].

### 4.1.4. Aromatic compounds

Measurements of aromatic compounds in Southern Hemisphere marine air are relatively scarce. Galbally et al. [35] detected 4 ± 1 ppt of benzene at Cape Grim; Lawson et al. [70] reported 10 ppt, 9 ppt, and 10 ppt each of benzene, toluene and $C_8H_{10}$ over the Chatham Rise in the south-west Pacific Ocean and Warneke and de Gouw [73] measured 9 ± 7 ppt of toluene in the north-west Indian Ocean. Colomb et al. [68] observed a latitudinal gradient for benzene between 24 and 49°S in the Indian Ocean. They reported a mole fraction of 20 ± 10 ppt for what they determined to be pristine marine air but observed 80 ± 40 ppt between 30°S and 40°S, which is the latitudinal range in which Wollongong is situated. The mole fractions observed in marine air reaching Wollongong for benzene (20 ± 10 ppt), toluene (30 ± 20 ppt), $C_8H_{10}$ (23 ± 6 ppt) and $C_9H_{12}$ (36 ± 7 ppt) are all within one standard deviation of each other, indicating photochemically aged air.

### 4.2. Terrestrial Biogenic VOCs during MUMBA

The left-hand column of Figure 4 shows modified wind roses for the entire campaign, where the length of each sector is proportional to the frequency of occurrence of winds from that direction and the wind roses are coloured by the range of mole fractions of different biogenic VOCs observed under those wind directions. The right-hand column of Figure 4 shows polar bivariate plots, where the shape represents the different wind speeds and directions that occurred during the campaign and the colour represents the mean mole fractions of the different VOCs observed under these wind conditions. The polar bivariate plots include the majority of the campaign days, but the extreme heat days (with strong westerly winds) have been removed, so that the patterns that represent the more usual conditions can be seen more clearly.

The first pair of plots in Figure 4 show that the highest mole fractions of isoprene are observed when winds are from the north-west (seen in the modified wind rose on the left-hand side). However, these winds are infrequent, and the general pattern visible in the polar bivariate plot—once the two hot days (8 and 18 January 2013) have been removed—is of higher mean mole fractions with winds from the east and the south-east, where Puckey's Estate is.

The same plots reveal a contrasting picture for the isoprene oxidation products (MACR + MVK). Although the highest mole fractions are also observed with winds from the north-west, the general pattern revealed by the polar bivariate plot is of very low mole fractions to the east and south-east, and of relatively high mole fractions to the north. This indicates two things:

1. the proximity of Puckey's Estate to the measurement site leaves little time for oxidation to take place, and
2. the escarpment is a source of isoprene; however, given its distance from the site (~10 km) and the average wind speed observed during the campaign (2.8 m s−1 or ~10 km/h), there is enough time (~1 hour) for oxidation of a substantial fraction of the isoprene to occur before the air reaches the site.

This difference in the extent to which oxidation occurs before isoprene reaches the measurement site depending on the source (and therefore on wind direction) is seen quite clearly in Figure 5, which plots isoprene versus the sum of isoprene and its oxidation products, coloured by wind direction. The solid black line is the 1:1 ratio, while the 2:1 and 1:2 ratios are indicated by the dotted black lines.

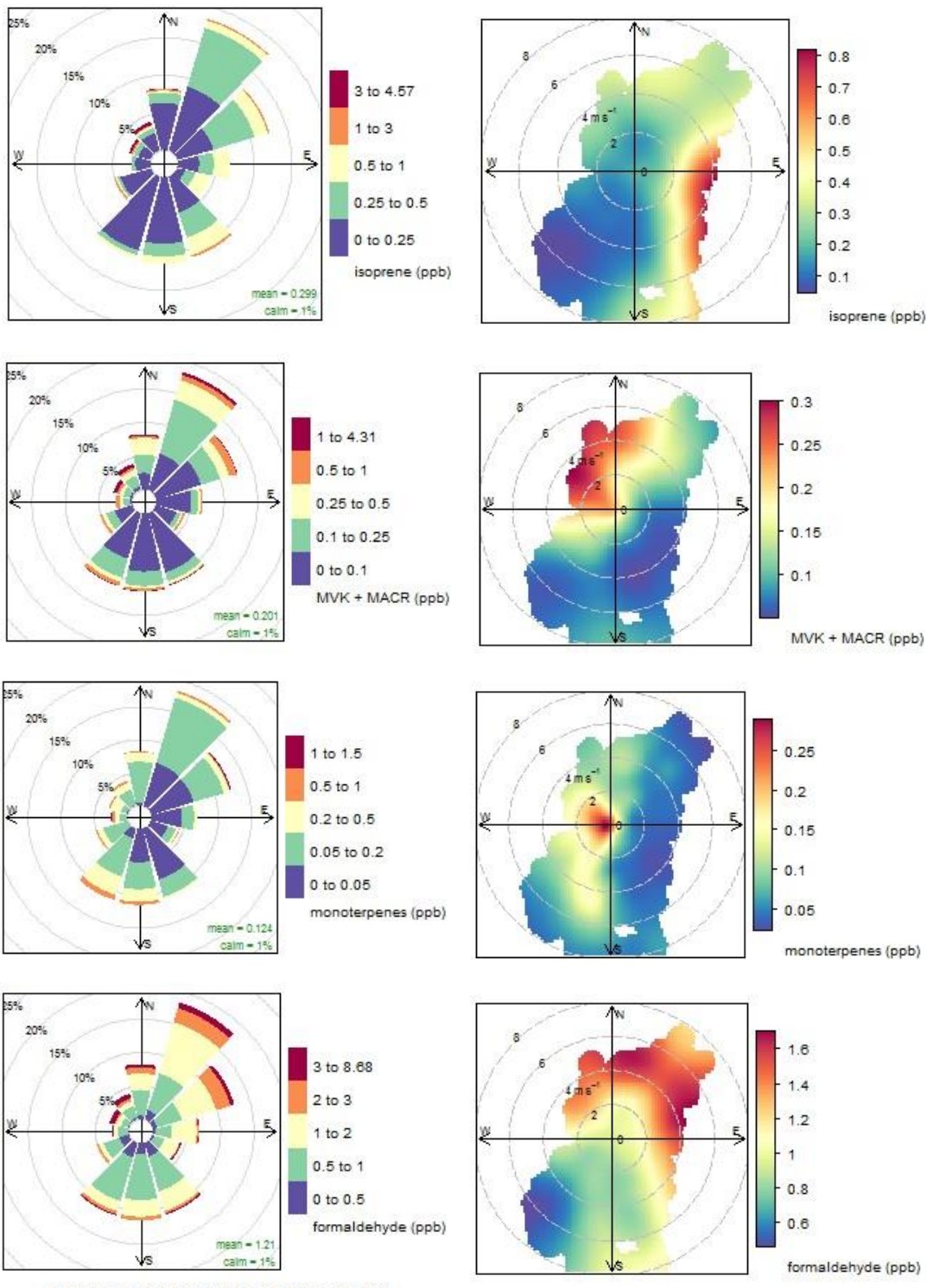

**Figure 4.** Modified wind roses for the whole campaign (left-hand side) and bivariate polar plots, with the extreme heat days removed (right-hand side) for isoprene, the sum of methacrolein and methyl vinyl ketone (MACR + MVK), monoterpenes and formaldehyde.

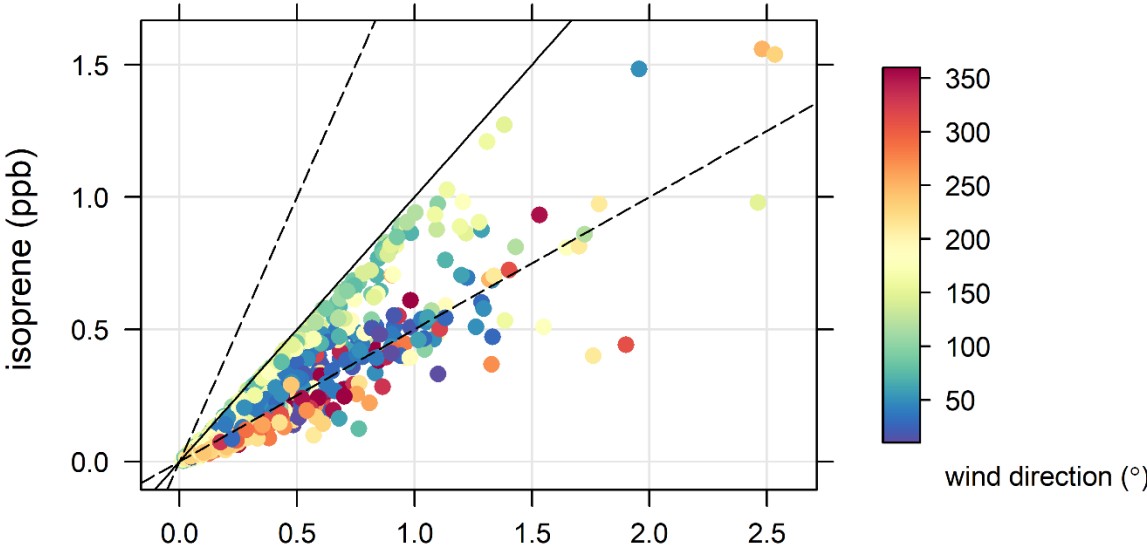

**Figure 5.** Isoprene vs. the sum of isoprene and its oxidation products (MACR + MVK), coloured by wind direction. The 2:1 and 1:2 ratios are shown as the dotted lines. Note that the two extreme heat days (8 and 18 January) are excluded from the data in this plot.

Figure 5 shows that when winds are from the east or south-east (green and yellow data points), isoprene makes up most of the sum of isoprene and its oxidation products, indicating very little oxidation. When winds are from the north and north-west (orange and dark blue dots), the oxidation products actually constitute the bulk of the sum of isoprene and its oxidation products, indicating extensive oxidation.

Additional potential source information is contained within the diel variations in observed mole fractions. Figure 6 shows a composite diel cycle for isoprene, built of box and whisker plots of observed values for each hour of the day. This composite diel cycle for isoprene shows the expected pattern of peak values in the middle of the day when maximum daylight and temperatures cause emissions to peak. Night-time mole fractions are close to zero as emissions cease at night fall and isoprene is short-lived.

Monoterpenes are emitted by vegetation both during the day and at night, with emissions depending mostly on temperature. Monoterpenes are very reactive species and therefore have short atmospheric lifetimes of a few hours [85], depending on atmospheric conditions. The composite diel cycle for monoterpenes during MUMBA shows peak amounts overnight and early morning and minimum amounts at midday (see Appendix B, Figure A3), reflecting the combined effects of increased loss processes for monoterpenes and dilution due to an increased boundary layer height during the day.

The modified wind rose and the bivariate polar plot for monoterpenes (third pair of plots in Figure 4) reveal a different pattern for monoterpenes to that of isoprene and its oxidation products. Whilst the highest mole fractions of monoterpenes are also observed from the west, they are observed mostly at low wind speeds. Weak westerlies are associated with katabatic drainage from the escarpment at night [39]. The escarpment therefore appears to be the main source of monoterpenes impacting the site, since very low mole fractions are observed coming from the east and south-east. This indicates that the vegetation in Puckey's Estate is a weak monoterpene emitter. It is also possible that monoterpenes are emitted from Puckey's Estate during the day, but do not reach the measurement site due to the high oxidation rates of monoterpenes in the afternoon; however, since Puckey's Estate is located so close to the measurement site, this scenario seems unlikely, especially given the observed pattern of isoprene oxidation products (MACR + MVK) discussed above.

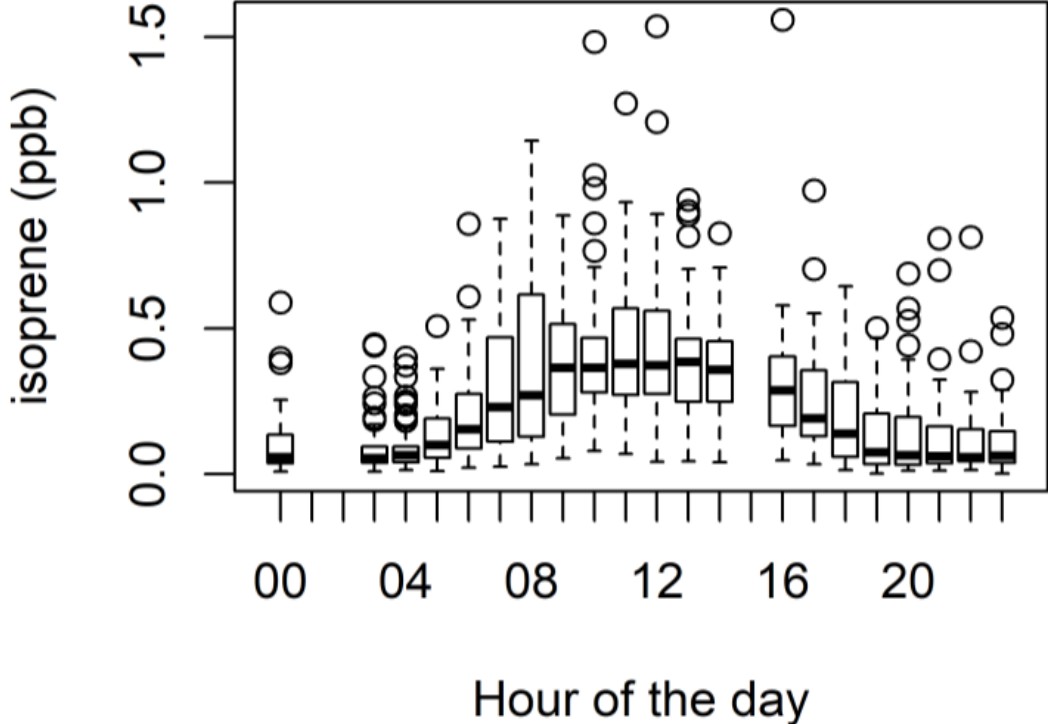

**Figure 6.** Composite diel cycle for isoprene, built of box and whisker plots showing the variability in observed mole fractions for each hour of the day. The thick black line is the median isoprene mole fraction observed, the box covers the 1st to the 3rd quantile, the whiskers are 1.7 times the interquantile range (IQR) and the dots represent values that fall outside of 1.7*IQR. Note that the two extreme heat days (8 and 18 January) are excluded from the data in this plot.

Formaldehyde is photochemically produced as part of VOC degradation processes. Its main losses are photolysis and reaction with OH as well as dry and wet deposition. The increasing mole fractions observed during the day at the MUMBA site indicate that production was generally greater than destruction (see diel cycle in Figure A3). During the night, losses due to dry deposition become more relevant in a shallower boundary layer.

The bottom pair of plots in Figure 4 show that formaldehyde has fairly diffuse sources. Its distribution resembles that of ozone, a trace gas that is exclusively photochemically produced. Formaldehyde precursor species can be of anthropogenic or biogenic origin. The highest observed mole fractions of formaldehyde are associated with the wind directions in the sector between the north-west via north and east. These winds are most prominent during the day, when photochemical production peaks. The lowest mole fractions are associated with winds from the south-west, which occur mostly at night.

Methanol, acetone and acetaldehyde are all longer-lived species, with lifetimes of a few days [78–80,86–89], and their diel cycles are similar (Figure A3). Methanol, acetone and acetaldehyde are all directly emitted by plants, including eucalypts (e.g., [78,90]). Methanol, acetone and acetaldehyde are also photochemically produced from precursors in the atmosphere [67,80,87] although methanol is primarily of biogenic origin. Methanol, acetone and acetaldehyde all show a peak in the early morning hours and minimum values in the late afternoon. This observed diel pattern is likely due to a combination of sources acting at night-time as well as daytime, a shallower boundary layer in the early morning, and the variation of the dominant wind direction being from the escarpment at night and the ocean during the afternoon.

The modified wind roses and polar bivariate plots for methanol, acetone and acetaldehyde in Figure 7 are also very similar, indicating common sources. This is corroborated by the strong correlation observed between these species and shown in Figure 8. The highest mole fractions of all

three species are observed when winds are from the north and the north-west, suggesting that the escarpment is a source of these compounds. The low mole fractions observed when winds are from the east or the south-east indicate that Puckey's Estate is a low emitter of these compounds, as well as a low monoterpenes emitter. The plots in Figure 7 also reveal slightly elevated mole fractions in the south-west and north-east quadrants for all compounds, with the north-eastern feature being more prominent for acetaldehyde.

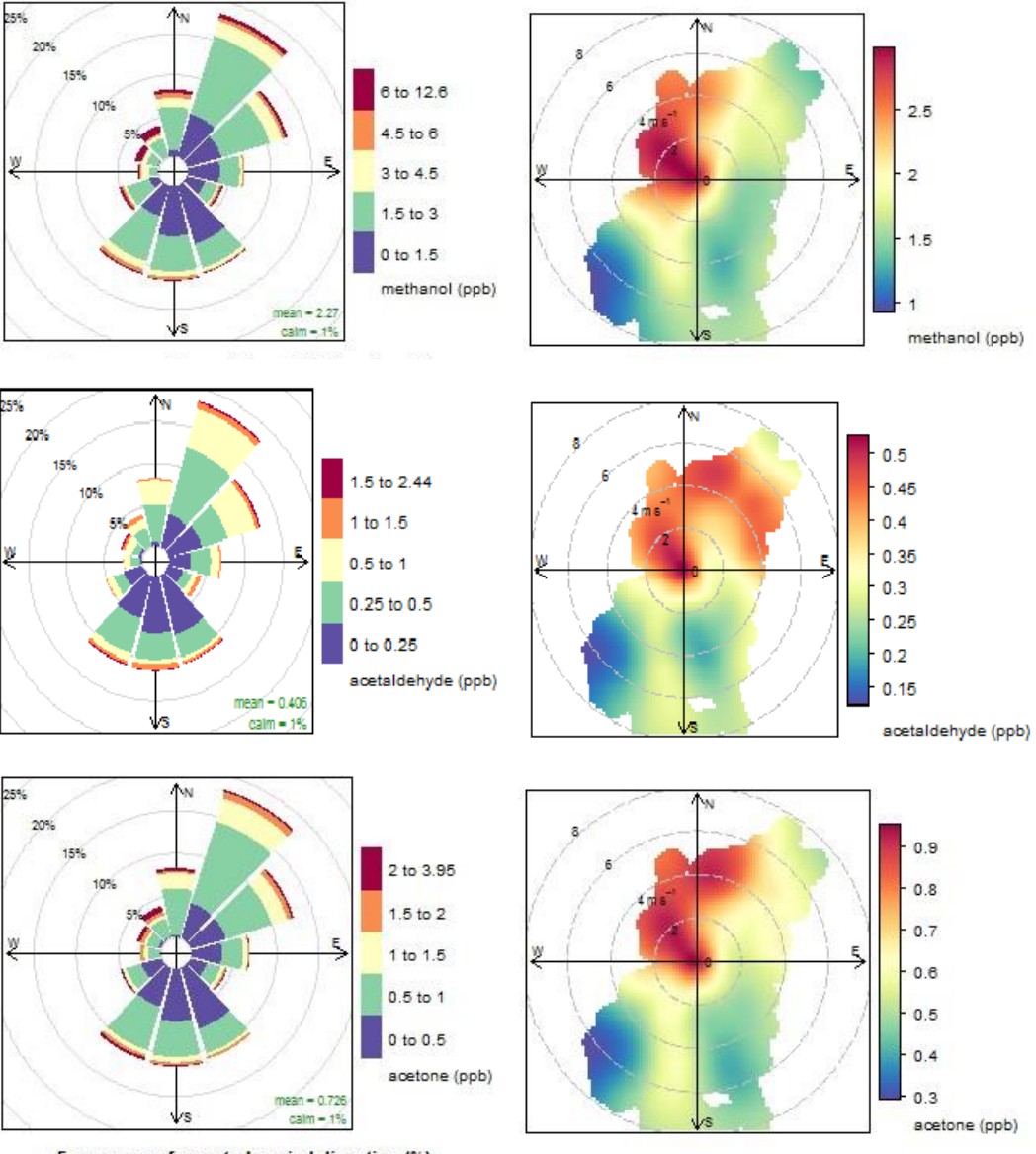

**Figure 7.** Modified wind roses for the whole campaign (left-hand side) and bivariate polar plots, with the extreme heat days removed (right-hand side) for methanol, acetaldehyde and acetone.

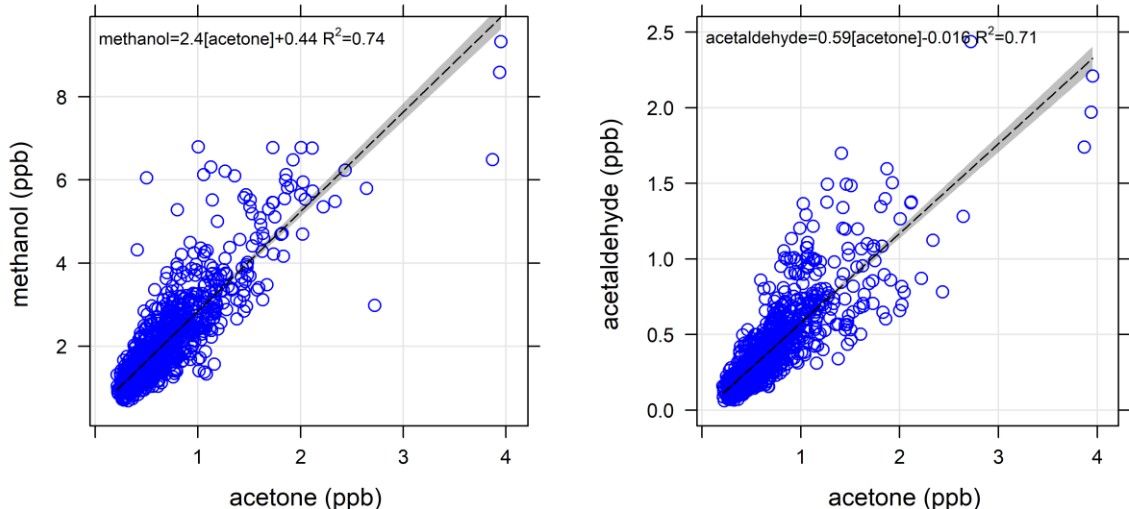

**Figure 8.** Scatter plots showing the strong correlations of methanol (**left**) and acetaldehyde (**right**) with acetone. These strong correlations indicate that these VOCs share common sources. Note that the two extreme heat days (8 and 18 January) are excluded from the data in this plot.

### 4.3. Characterisation of Terrestrial Biogenic VOCs on Atypically Hot Days

Although not record breaking, the temperatures experienced on 8 January and 18 January 2013 are >15 °C hotter than the long-term daily average maximum temperature for January at Bellambi AWS (24.8 °C). As this type of weather might become more frequent under a warmer climate, measurements made on 8 January and 18 January may hold clues as to how this would affect the region in terms of VOC emissions and air quality.

The highest total VOC loadings (based on the sum of the twelve quantified VOCs) were observed on the hot days. These total loadings were dominated by oxygenated VOCs of biogenic origin. The highest ozone mole fractions of the campaign were also observed on those days. These ozone events are discussed in more detail in Utembe et al. [28].

Figure 9 shows the evolution of some meteorological conditions and BVOC mole fractions over two 3-day periods encompassing each of the extreme heat days. The plot on the left shows wind direction, wind speed, temperature and some BVOC mole fractions for 7 January until 9 January 2013 and the plot on the right shows the same for the period of 17 January until 19 January 2013. The days on either side of 8 January and 18 January are fairly typical and representative of the MUMBA campaign, and are shown to provide context for the atypical values observed on the extreme heat days. The gaps in the BVOC records are due to ion source issues on 7 January and to the shutdown of the PTR-MS on 18 January.

Both hot days were characterised by strong north-westerly winds and very high temperatures. On both days, low pressure systems drew hot dry air from inland and sea breezes did not develop. On 8 January, north-westerly winds persisted for 16 h, from 06:00 until 22:00, at which time the hot spell broke suddenly, with wind changing to a southerly direction and the temperature dropping by just over 9 degrees within a single hour. The highest mole fractions of formaldehyde, and some of the highest mole fractions of isoprene, its oxidation products (MACR + MVK) and methanol were observed on that day. Isoprene mole fractions increased from 08:00 and remained high until 18:00, when light levels started to decline. Formaldehyde and the isoprene oxidation products increased from 08:00 and peaked at 10:00, then slowly decreased over the rest of the day. In contrast, methanol increased as the wind shifted direction just before 06:00 and remained relatively constant (>8 ppb) all day until the southerly change at 22:00. The same pattern was observed for monoterpenes, acetone and acetaldehyde (not shown). These species were not present in unusually high amounts, but they remained at consistently high mole fractions during the entire north-westerly spell. The sharp gradient

in mole fractions observed with the change in wind direction indicates transport of these species from forested areas north-west of Wollongong.

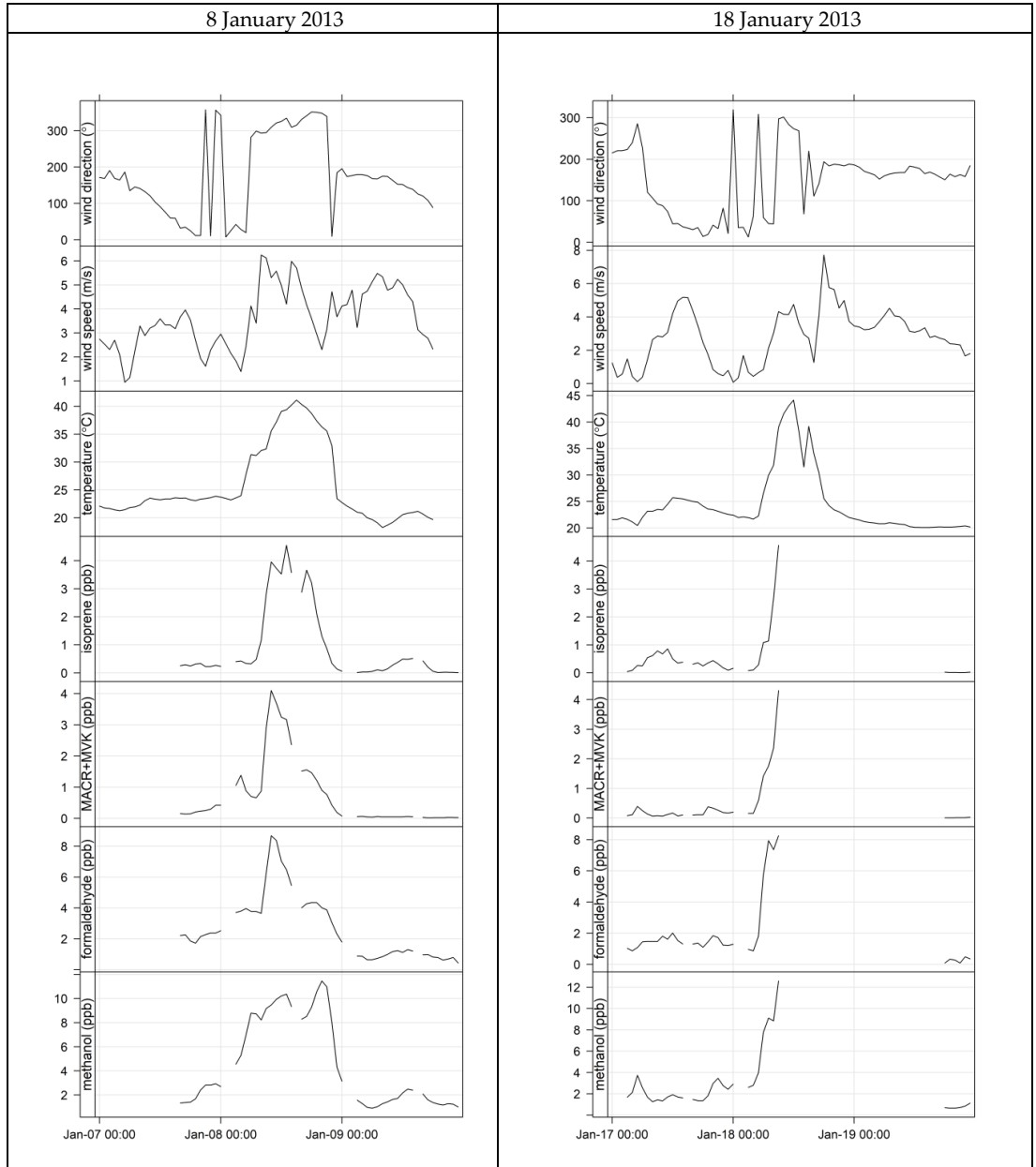

**Figure 9.** Evolution of meteorological conditions and selected VOCs over three-day periods encompassing 8 January (**left**), and 18 January (**right**). The gaps in the VOC record are due to ion source issues on 7 January and to the shutdown of the PTR-MS on 18 January.

On 18 January, the north-westerly winds started later, and ended sooner (from 09:00 until 14:00). From 14:00, the wind direction was variable, until a strong southerly change happened from 18:00. On that day, the temperature increased by over 7 degrees between 08:00 and 09:00, coinciding with the shift to north-westerly winds. During the same period, isoprene jumped nearly 2 ppb to reach the highest hourly-averaged mole fraction recorded during the campaign. There is no VOC data for the remainder of that day, because most of the instruments, including the PTR-MS, had to be turned off due to inadequate air conditioning.

Although the absolute amount of VOCs observed on a given day is of some relevance, it is generally more useful to compare ratios of various species. Figure A4 shows the same plot as Figure 5,

but including the data from the hot days. It indicates that the air reaching the site on those days was photochemically aged, since the ratio of isoprene to the sum of isoprene and its oxidation products is less than 1. Figure A5 show the same correlations as in Figure 8, but with the data from the hot days plotted in red. The plots indicate that the methanol to acetone ratio may be higher on the hot days than during the rest of the campaign, but that the acetaldehyde to acetone ratio is unchanged.

Other ratios of interest include the ratio of isoprene to monoterpenes as it may have implications for secondary organic particle formation, with isoprene thought to inhibit new particle formation events [91,92]. In the top panel of Figure 10 monoterpenes mole fractions are plotted against isoprene and coloured by temperature to highlight the data collected on the hot days. The darker blue data points are mostly night time data, as temperatures tend to be lower at night than during the day. There is little isoprene at night, and the monoterpenes to isoprene ratio is high. During the day (lighter blue points), the monoterpenes to isoprene ratio is low. On the hot days (red data points), the mole fractions of isoprene are much higher than on other days, but the observed monoterpenes mole fractions are not unusually high, resulting in a lower than usually observed monoterpene to isoprene ratio.

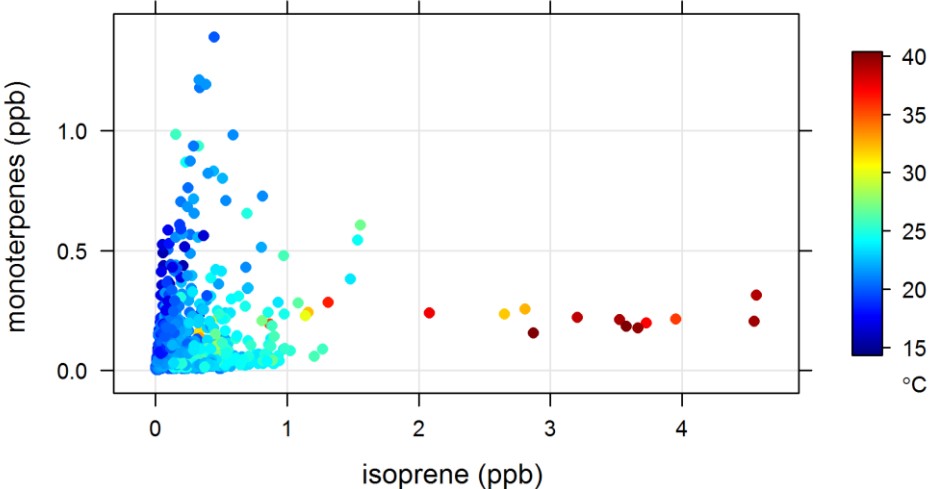

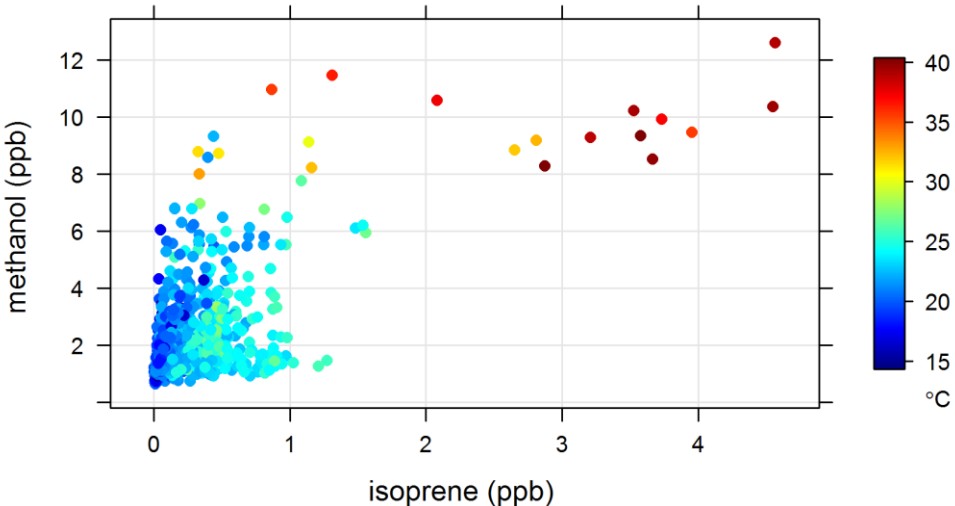

**Figure 10.** Monoterpenes (**top panel**) and methanol (**bottom panel**) plotted against isoprene and. coloured by temperature to highlight the hot days.

The bottom panel of Figure 10 shows that both isoprene and methanol mole fractions are very significantly greater on the hot days than on other days. However, for isoprene between 2 and 5 ppb, on the hot days, the methanol levels do not increase in parallel with the isoprene increase. We suggest that this arises from a methanol level in air from the forested escarpment that is significantly higher than that in marine air that has passed over the vegetation in Puckey's estate, consistent with our understanding of the atmospheric methanol cycle [78]. Thus, unlike isoprene, it appears likely that plant methanol emissions do not increase significantly as a result of the extreme heat.

## 5. Conclusions

The MUMBA campaign has provided a rich dataset for exploring the complex chemistry of the atmosphere in the marine/urban/forest interface. During the 8-week campaign the MUMBA site experienced some very different conditions, ranging from unpolluted marine air with composition akin to that measured under baseline conditions at Cape Grim, Tasmania to highly polluted air influenced by local urban pollution from nearby steelworks [39]. Many VOC species were measured at the MUMBA site during the clean marine air episode on 26 December 2012. These measurements provide useful information about background concentrations in clean marine air at these latitudes. Typical unpolluted marine air mole fractions during summer 2012–2013 at this latitude (34°S) were established for $CO_2$ (391.0 ± 0.6 ppm), $CH_4$ (1760.1 ± 0.4 ppb), $N_2O$ (325.04 ± 0.08 ppb), CO (52.4 ± 1.7 ppb), $O_3$ (20.5 ± 1.1 ppb), acetaldehyde (190 ± 40 ppt), acetone (260 ± 30 ppt), dimethyl sulphide (50 ± 10 ppt), benzene (20 ± 10 ppt), toluene (30 ± 20 ppt), $C_8H_{10}$ aromatics (23 ± 6 ppt) and $C_9H_{12}$ aromatics (36 ± 7 ppt). We conclude that clean marine air sampled during the MUMBA campaign provides a useful constraint on background concentrations of a number of trace gases of interest and may supplement information provided by distant WMO GAW stations.

The MUMBA campaign also provided interesting observations of biogenic VOCs, which predominantly originated from the forested escarpment to the north-west of the measurement site, except for isoprene, which predominantly came from a nearby forested strip of parkland to the east. Wollongong experienced two of its hottest days on record during the MUMBA campaign, with mean temperatures in excess of 40°C for several hours. Both extreme heat days had strong westerly winds that brought high biogenic signals from the forested escarpment to the west of the measurement site. The mixture of biogenic VOCs observed with winds from the escarpment on the extreme heat days differed from that observed from the nearby forested region during the rest of the campaign, but whether this was due to the extreme temperatures or from differences in emissions from the different vegetation present is unclear. More comprehensive measurements of biogenic VOCs are needed, including how the atmospheric abundances of these species changes in different biomes and during different seasons of the year.

**Author Contributions:** Conceptualization, C.P.W., E.A.G., I.G., M.K. and R.H.; Methodology, C.P.W., E.A.G., D.K. and I.G.; Software: S.M.; Validation, E.A.G., D.K., D.W.T.G. and S.Z.C.; Formal Analysis, E.A.G. and C.P.W.; Investigation, E.A.G., C.P.W., D.K., R.H., S.W., D.D., R.B., I.G., P.K., S.L., Z.L., R.L.L., S.M., P.S., A.G. and S.Z.C.; Data Curation, E.A.G., D.D. and S.M.; Writing—Original Draft Preparation, C.P.W. and E.A.G.; Writing—Review & Editing, all authors.; Visualization, E.A.G.; Supervision, C.P.W., S.R.W. and I.G.; Project Administration, C.P.W.; Funding Acquisition, C.P.W. and I.G.

**Funding:** This research was funded in part by Australia's National Environmental Science Program through the Clean Air and Urban Landscapes hub and from the Australian Research Council Discovery Project DP160101598. This research was also supported by Australian Government Research Training Program (RTP) Scholarships. The participation of I.G., S.L. and S.M., along with the PTR-MS, auxiliary equipment and provision of calibration gases was funded through CSIRO Project R-03340-01 VOC Strategic Development 2012–2013.

**Acknowledgments:** The authors acknowledge the NSW Office of Environment and heritage for providing publicly available air quality data in Wollongong. The authors would like to thank all those from the University of Wollongong's Centre for Atmospheric Chemistry and CSIRO's Climate Science Centre group who helped with the logistics of undertaking an extensive measurement campaign. Thanks are also due to Kids Uni & the Science Centre for their helpful support during the campaign. The analyses presented here were carried out using the statistical software R [93] and the openair package [94].

**Conflicts of Interest:** The authors declare no conflict of interest.

## Appendix A

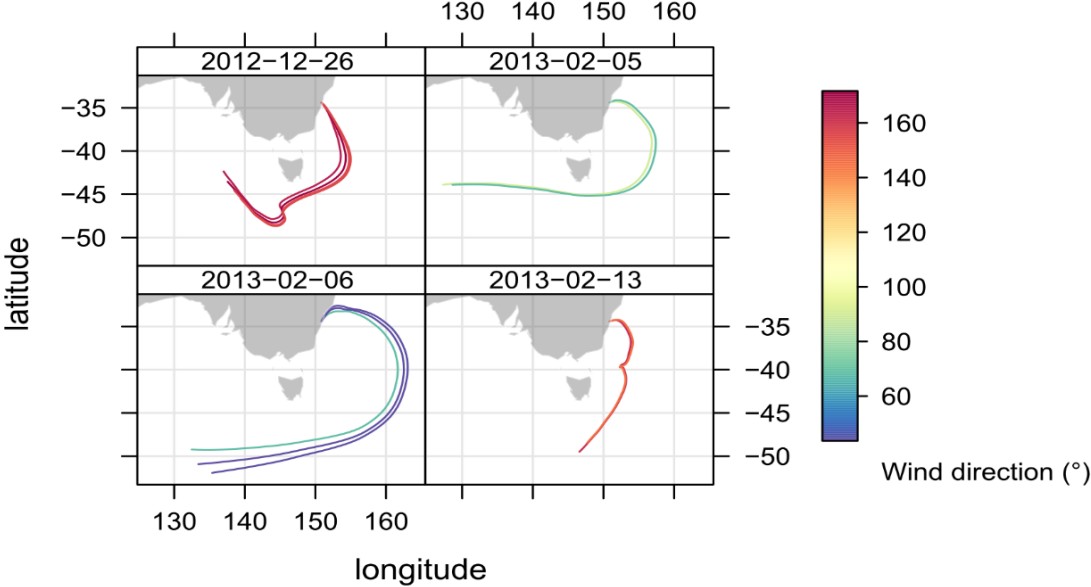

**Figure A1.** Pre-calculated back-trajectories are available for Wollongong every three hours through the R package openair [94]. These trajectories are calculated using HYSPLIT (Hybrid Single Particle Lagrangian Integrated Trajectory Model) [95] from a height of 10 m and propagated backwards in time for 96 h. This figure shows the 96-h back-trajectories whose arrival times coincided with periods of low radon (<200 mBq/m$^3$), coloured by the wind direction observed in Wollongong at the time of arrival. Arrival times are 09:00, 12:00, 15:00 and 18:00 AEST on 26 December 2012, 12:00 and 15:00 AEST on 5 February 2013, 12:00, 15:00 and 18:00 on 6 February 2012 and 15:00 AEST on 13 February 2013.

**Table A1.** Mean mole fractions for thirteen VOC species, CO, NO$_x$ and ozone, observed over periods during which marine air reached the MUMBA site. The standard deviation is provided to illustrate variability in the mole fractions over those periods and does not reflect measurement error. All mole fractions are in ppt unless otherwise indicated.

| Species | South-Easterly Winds | | North-Easterly Winds | |
| --- | --- | --- | --- | --- |
| | December 26th 08:00–18:59 | February 13th 14:00–17:59 | February 5th 12:00–17:59 | February 6th 13:00–18:59 |
| Formaldehyde | 590 ± 80 | 600 ± 130 | 770 ± 170 | 910 ± 390 |
| Methanol | 1340 ± 170 | 1020 ± 70 | 1190 ± 110 | 1120 ± 170 |
| Acetonitrile | 56± 5 | 65 ± 4 | 66 ± 3 | 70 ± 4 |
| Acetaldehyde | 190 ± 40 | 120 ± 30 | 200 ± 40 | 170 ± 40 |
| Acetone | 260 ± 30 | 270 ± 10 | 320 ± 20 | 330 ± 30 |
| Dimethyl sulphide | 50 ± 10 | 35 ± 8 | 38 ± 8 | 40 ± 20 |
| Isoprene | 370 ± 120 | 410 ± 30 | 380 ± 200 | 160 ± 30 |
| MACR + MVK | 40 ± 10 | 39 ± 9 | 54 ± 8 | 46 ± 7 |
| benzene | 20 ± 30 | 28 ± 5 | 40 ± 20 | 50 ± 20 |
| toluene | 30 ± 20 | 45 ± 3 | 90 ± 70 | 180 ± 170 |
| C$_8$H$_{10}$ | 23 ± 6 | 70 ± 40 | 80 ± 40 | 90 ± 30 |
| C$_9$H$_{12}$ | 36 ± 7 | 50 ± 20 | 60 ± 20 | 90 ± 30 |
| monoterpenes | 33 ± 15 | 25 ± 9 | 23 ± 5 | 18 ± 6 |
| CO (ppb) | 52 ± 2 | 58 ± 6 | 60 ± 10 | 66 ± 4 |
| NO$_x$ | 990 ± 210 | 1850 ± 220 | 3300 ± 1500 | 3700 ± 690 |
| Ozone (ppb) | 20.5 ± 1.1 | 20.3 ± 0.8 | 18 ± 2 | 22 ± 3 |

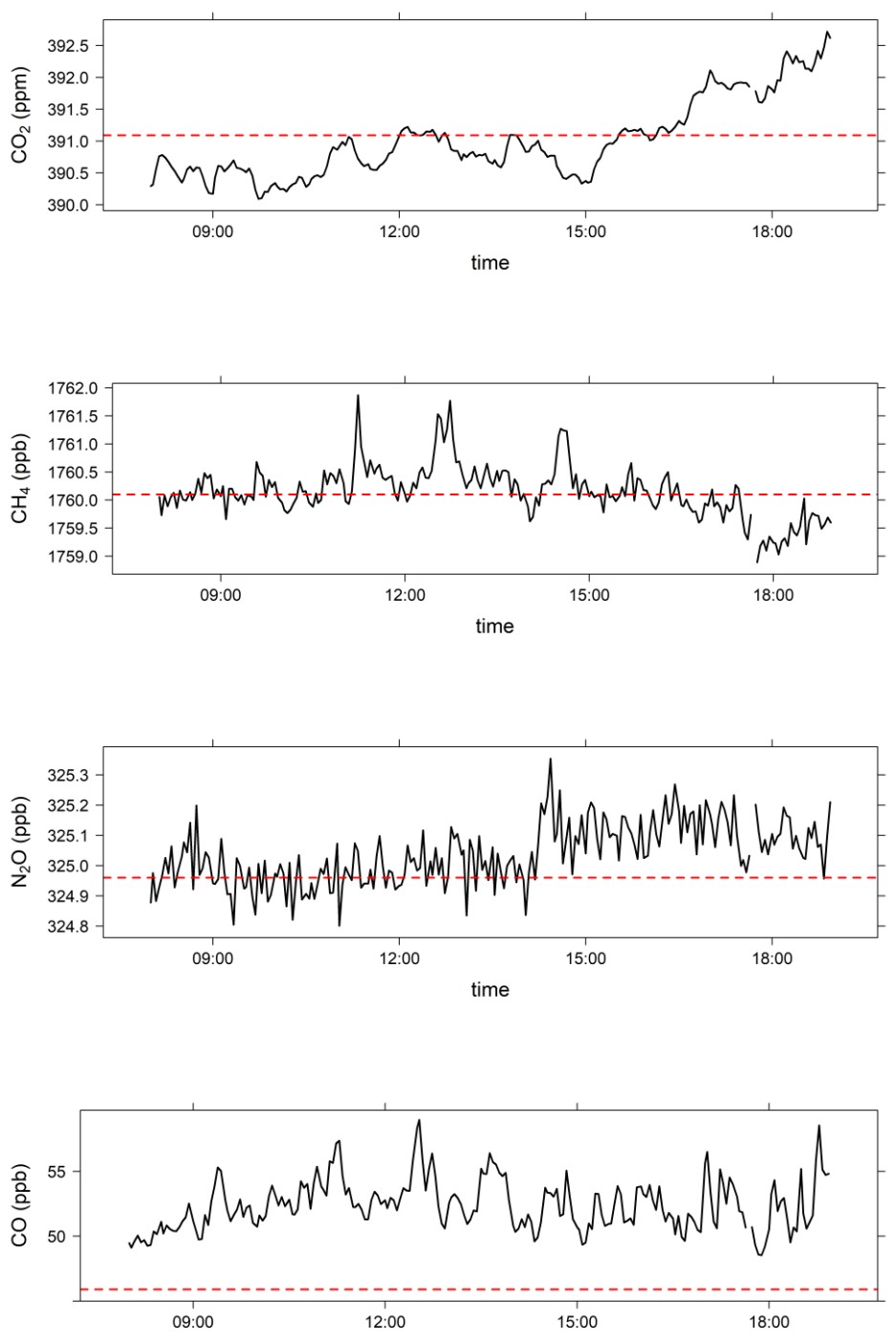

**Figure A2.** 3-min timeseries of $CO_2$, $CH_4$, $N_2O$ and $CO$ on 26 December 2012 between 08:00 and 18:59. The red dashed horizontal line is the Cape Grim baseline value for 26 December 2012.

## Appendix B

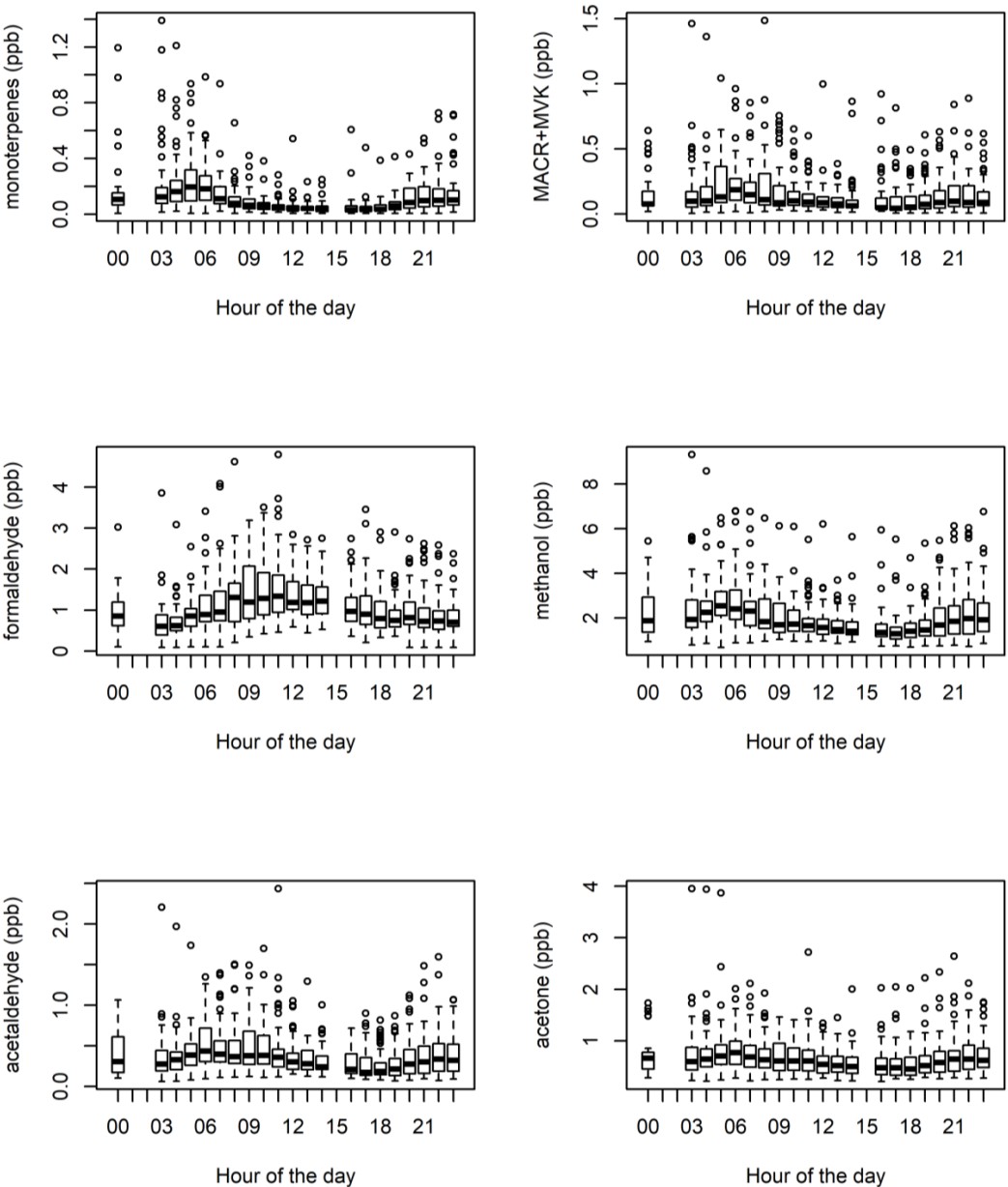

**Figure A3.** Composite diel cycle for monoterpenes, the sum of methacrolein and methyl vinyl ketone (MACR + MVK), formaldehyde, methanol, acetaldehyde and acetone, built of box and whisker plots showing the variability in observed mole fractions for each hour of the day. The thick black line is the median mole fraction observed, the box covers the 1st to the 3rd quantile, the whiskers are 1.7 times the interquantile range (IQR) and the dots represent values that fall outside of 1.7*IQR. Note that the two extreme heat days (8 and 18 January 2013) are excluded from the data in this plot.

**Appendix C**

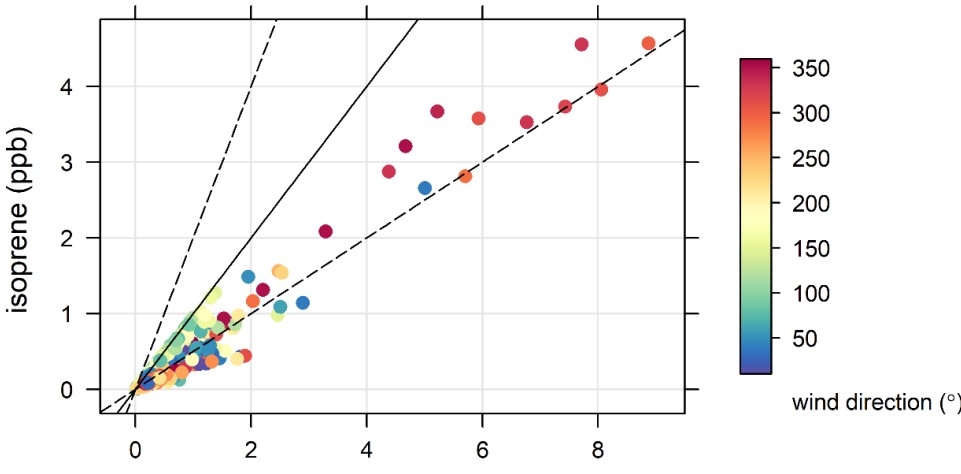

**Figure A4.** As in Figure 5, but including data from the hot days.

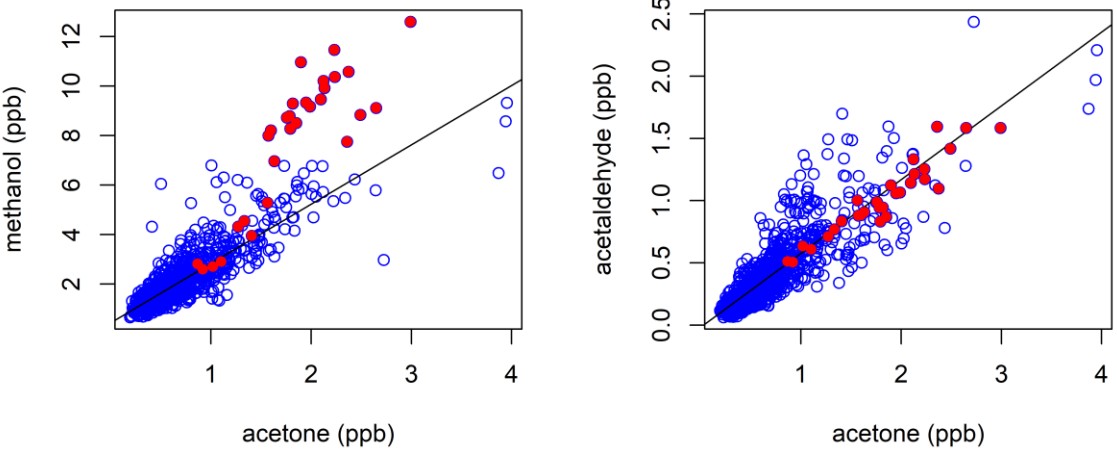

**Figure A5.** As in Figure 8, but with the hot days data included in red.

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
