# Peer review of "Composition of Clean Marine Air and Biogenic Influences on VOCs during the MUMBA Campaign"

_atmosphere, doi:10.3390/atmos10070383_

Round 1

Reviewer 1 Report

I'd like to thank authors for responding to the review. I think the major improvement is the clarification of the aims of the publication and focus on the non-urban wind sectors and the improvements in the presentation of the data. However, the paper does still report and discuss terrestrial origin measurements and some acknowledgement of the great deal of literature that exists on biogenic emissions (e.g. UC Berkeley forest measurements) would have been beneficial in additional to a discussion of other global measurements of mixed terrestrial biogenic/marine influenced VOC measures such as the MPIC Cyphex measurements.

I would still like to see more detail on the inlet losses, i.e. how was the test done, was the inlet changed during the campaign, how long was it, comment on the contamination of the inlet my urban wind sectors during clean periods by re-volatilization of adsorbed components or at least an acknowledgement that these things are a source of error. 5% losses of a large concentration are can be a large source of contamination during "clean air" periods.

Author Response

We thank the reviewer for taking the time to read our manuscript again. It is much apppreciated.

Reviewer 2 Report

The authors have addressed nearly all of my original concerns. However, I would like a couple clarifications.

1) The authors note that on Dec 26, there is a 10-hr period where temperatures changes by less than a degree centigrade, and other meteorological parameters (wind speed, direction, humidity) also are nearly constant. This seems unrealistic, but maybe not. For temperature to remain constant for 10 hours implies very cloudy conditions? If in fact wind speed, directions and temperatures were constant for the entire 11 hour period, please state this explicitly.

2) It appears the authors have not fully addressed my concerns about trajectories raised in my first review. Fig A1 shows three trajectories on Dec 26. Why three? Please clarify this figure and caption it appropriately. For some ambiguous reason, the trajectories are color-coded based on wind direction? Wind direction at what time? (the end of the trajectory?) Are these at three different heights in the PBL? or three different times during the 11-hour period? Furthermore, the caption for Fig. A1 mentions trajectories are calculated at 3-hr intervals, Does that mean that a new trajectory is calculated ending at the measurement site location every three hours? or that the time step of the back trajectory of each individual trajectory is 3 hours? During an entire 11 hour period, the authors MUYST calculate all back trajectories encompassing the entire time of the background measurement period, at least every three hours, and at least at three heights in the base, middle and top of the PBL. Therefore, for the 11-hr period on Dec 26, I would like to see trajectories calculated arriving near the beginning, middle and end of the 11-hr period at Wollongong, and calculated backwards from a height of 10, 200, and 1000m for all of those three times, for a total of nine trajectories.

Author Response

We thank the reviewer for taking the time to read our manuscript again. 

Reviewer 3 Report

I accept the authors' explanations and corrections in the textThe quality of the manuscript  has improved significantly.

Author Response

We thank the reviewer again for taking the time to review our manuscript and for providing helpful suggestions that have led to an improved paper.

This manuscript is a resubmission of an earlier submission. The following is a list of the peer review reports and author responses from that submission.

Round 1

Reviewer 1 Report

Review of Guerette et al., Composition of clean marine air and biogenic 3 influences on VOCs during the MUMBA campaign.

This paper is a description of the VOC data measured using PTRMS and FTIR during the “Measurements of Urban, Marine and Biogenic Air” (MUMBA) campaign in austral summer 2012-2013.

Overall comments.

This data contributes to filling a genuine gap in the data collection in the region. The paper is in the main a description of the data and analysis with wind sector and back-trajectory. The objectives of the study could be clearer and to the point. It is part of series of four papers that describe the campaign and to a degree this dilutes the utility of the paper. It is also not clear how 18 authors are responsible for met data and two instruments as the authorship for experimental planning and logistics will have been included in the overview paper.

As this is an urban study of VOCs it would be good to see far more acknowledgment and discussion of comparable studies in the literature in order to put these measurements in context. There have been many urban studies of VOCs very few of them are cited here and this is a large omission. The few that are cited are mentioned in the introduction as previous studies without further discussion and analysis. This contextual discussion would add for more the publication.

The main conclusion of the paper are that the campaign sampled different wind sectors and that the results can be useful for modelling. It is not hard to think that some further analysis of this dataset would do far more justice to the effort and resources that were very likely put into the experiment.

Specific comments

The abstract lists the average composition of the trace gases measured during the experiment and vague comments about the forested wind sectors. I would suggest that the authors are more terse and specific, i.e. mention the dominant species, be more specific about how the BVOC signature changes with source region. Currently the abstracts doesn’t want me to read the rest of the paper.

The description of PTRMS is vague. Which instrument is used, who is the manufacturer, what is the detector? A lot of this can be inferred but I shouldn’t have to infer it. How were losses of BVOCs in the inlet line characterised? There will be inlet boas due to losses; were these quantified? Was inlet heated to a constant temperature? All of this information should be dealt with in the manuscript.

How were the back trajectories calculated? Refer to partner paper when first describing the use in depth and briefly describe the salient information.

A substantial amount of data is presented and it’s difficult to take it all in. I would suggest some further data reduction. What to the mean diel cycles look like for the binned data? Why are the average diels in the appendix? Their picture quality makes it difficult to read. From each binned source region? Use your air mass origin classification to segment the time series presented. Figures 2, 4 and 5 are currently not adding much other than showing a time series. Could you block by air mass origin?

In figure 3, how are the fits calculated?

Section 4.1 seems to be nearly exclusively a description of previous work with very little additional analysis arising from the measurements that were already described in the dataset paper *reference 32). Why was no modelling done? What is the link between BVOC impacts on ozone formation in conjunction with local NOx production? Why is there no coherent story threading through the work published from this experiment? Obviously that’s because Paton-Walsh et al. has already partly done this which leaves very little scope for analysis with this dataset.

Very minor issues: there are a number in-line cites unformatted. And the dataset has not been updated to the published version.

Overall this paper adds very little to the analysis of the experimental data in what seems to have been a comprehensive field experiment. Having read most of the partner papers, it seems that a trick was missed here with further analysis looking at the proportions of AQ events and the effect BVC impacts on AQ with recommendations for reduction. Paton-Walsh state that the air shed is VOC limited. Then the need to for a more in depth analysis of the proportion of photochemical oxidation products produced by non-urban air masses.

Author Response

We thank Reviewer 1 for taking the time to review our manuscript.

Point-by-point responses are attached.

Reviewer 2 Report

Summary- The paper presents VOC measurements performed over several week's periods at a location in Coastal Australia. The measurements are sound and this is an area where explicit spectated measurements like this are needed, especially in remote locations like this. I recommend publication after minor revisions, where the authors carefully define the time periods averaged for all the measurements presented in tables and figures.

Particular points:

1) On  page 5, line 194: I have never been a fan of complex or even simple trajectory analyses in studies like this, and the authors could totally remove this trajectory analysis section and replace it with simple statements about wind speeds and directions during the sampling period. However, if the authors keep this trajectory analysis, it must be improved significantly. Why only three trajectories? What height are these trajectories calculated for? The authors must calculate trajectories from ALL (at least top, middle, bottom of PBL) layers in the PBL for ALL hours of the 17-hr period they are discussing to demonstrate "clean" air. I'm pretty sure more trajectories (from other levels and times) overlaid on this figure would look like spaghetti with trajectories coming from nearly everywhere. So maybe the authors could show that back trajectory clouds 'have a tendency' to spend more time over remote areas compared to other time periods. Radon is probably a much better indicator of continental influence if that is what the authors are trying to show. Even water vapor or dew point would provide "evidence" of marine vs. continental influences.

2) On pg 6, line 229: The authors are removing 10-minute 'spikes' in toluene and/or acetone from some of the longer-term averages presented. Are the authors only removing the toluene acetone spikes, or are toluene and acetone used to screen all of the other measurements presented? If so, why those particular species?, and what thresholds are used to objectively define a 'spike"? and is the spike required for BOTH or EITHER? Why shouldn't spikes in CO of NOx be removed? The whole concept of ignoring certain measurements because they are outliers could be scientifically unsound, and I would recommend showing us all the data unless there is something that sounds logical and objective about ignoring perfectly good measurements. At a minimum, show us "raw" and "screened" data in tables and figures.

3) On page 6 references to Table 2: Please clearly state time averaging period of measurements. This same comments applies to all figures and tables where concentrations are presented. I notice in the appendix, there are 'box-whisker' diagrams of what I assume are 10-minute 'raw' measurements then binned and averaged up to hourly averages. 

4) On pg. 7, there is a references to Fig A1. What do the colorings on Fig A1 mean? (eg CO<50ppb?)

Author Response

We thank Reviewer 2 for taking the time to review our manuscript.

Point-by-point responses are attached.

Reviewer 3 Report

The paper is very interesting, well written and organized, and represents some progress in understanding of background  concentrations of VOCs in the clean atmosphere near Australia which  may be a reference to concentration levels in other parts of the world, particularly to more polluted regions in the Northern Hemisphere. To the  authors’ knowledge, there is also  little data available on this region and in this sense, the paper  could be  valuable as it characterizes marine air masses in Southern Hemisphere midlatitudes.  The experimental section and the methodologies are adequate and clearly presented except for one issue, i.e. trajectory analysis.  The  results  are thoroughly discussed and the conclusions  show that the objectives of the study  have been met, i.e. both concentrations of  VOCs in  clear  marine air  and  VOCs of biogenic origin have been presented.  It is a pity that only a small number of cases were analyzed.   In addition,  I have some minor  comments that should be addressed:

1.      Please attach a map with the location of  the measurement point  and indicating potential sources of VOC biogenic emissions

2.      In Table 1 the detection limits are for a 1-hour measurement – Were all  concentrations  averaged to 1 hour  (Table 3, 4, fig. 3,4 , etc.)? Please give averaging times, because they are different, 3 min, 10 min, 1-h, etc.

3.      P lease provide the source of the trajectory analysis (also the height, frequency).

4.      l. 264-267 -  “  This  might be attributed to local traffic on the road east of the site. The higher background may be also due to local influences (shipping, steel works), ......”  - this interpretation is inconsistent with the statement that this day  concerned  the  clear air episode.

5.      l. 285 {Singh, 2001 #359;Singh, 2004 #304}  - please give the number from the bibliography.

6.      l. 308 – “....four species show peak amounts around 05:00 06:00 AEST “it does not follow from the data in Figure  2.

7.      Figs 2,4,6 - I am not convinced if it is necessary to give information in the form of colours, it results from drawings.

Author Response

We thank Reviewer 3 for taking the time to review our manuscript.

Point-by-point responses are attached.
